# DIVA-GRPO: Enhancing Multimodal Reasoning through Difficulty-Adaptive Variant Advantage

**Haowen Gao**[1,2†]**, Zhenyu Zhang**[3†]**, Liang Pang**[1*]**, Fangda Guo**[1]**, Hongjian Dou**[3]**,
Guannan Lv**[3]**, Shaoguo Liu**[3*]**, Tingting Gao**[3]**, Huawei Shen**[1]**, Xueqi Cheng**[1]

[1] State Key Laboratory of AI Safety, Institute of Computing Technology, CAS, Beijing, China
[2] University of Chinese Academy of Sciences, Beijing, China
[3] Kuaishou Technology, Beijing, China

`gaohaowen23s@ict.ac.cn, zhangzhenyu06@kuaishou.com`
`pangliang@ict.ac.cn`

## Abstract

Reinforcement learning (RL) with group relative policy optimization (GRPO) has become a widely adopted approach for enhancing the reasoning capabilities of multimodal large language models (MLLMs). While GRPO enables long-chain reasoning without a traditional critic model, it often suffers from sparse rewards, arising from the scarcity of positive feedback on difficult problems, and from advantage vanishing, which occurs when group-level rewards exhibit high consistency for problems that are too easy or too hard. Existing solutions fall into three categories: sample enhancement and expansion, which may aggravate vanishing advantage due to poor control of difficulty distribution; selective sample utilization, which fails to fully leverage the value of all data; and indirect reward design, which may introduce biased optimization directions due to misalignment between reasoning and the final outcome. However, these approaches overlook a fundamental question: ***for a given problem, how can we ensure that the within-group reward distribution of responses exhibits enough variance to yield clear optimization signals for each response?*** To address these issues, we propose DIVA-GRPO, a difficulty-adaptive variant augmentation advantage method that dynamically adjusts the difficulty distribution of variants for each problem from a global perspective. Our method dynamically assesses problem difficulty, samples variants with appropriate difficulty levels, and calculates advantages within both local and global (a problem and its variants) groups using difficulty-weighted and normalized scaling. This design alleviates reward sparsity and advantage vanishing, minimizes data waste, and improves training stability. Extensive experiments on six reasoning benchmarks demonstrate that DIVA-GRPO outperforms existing approaches in both training efficiency and reasoning performance. Code is available at `https://github.com/Siaaaaaa1/DIVA-GRPO`.

## 1 Introduction

Multimodal large language models (MLLMs) (Chen et al., 2024b; Hurst et al., 2024; Laurençon et al., 2024; Liu et al., 2023; Yin et al., 2024; Wu et al., 2024b; Alayrac et al., 2022; Zhu et al., 2023; Li et al., 2023) have demonstrated remarkable ability to integrate textual and visual information (Pang et al., 2024) for complex reasoning tasks, such as visual question answering (Antol et al., 2015; Xiao et al.,

---

[†]Equal contribution
[*]Corresponding authors

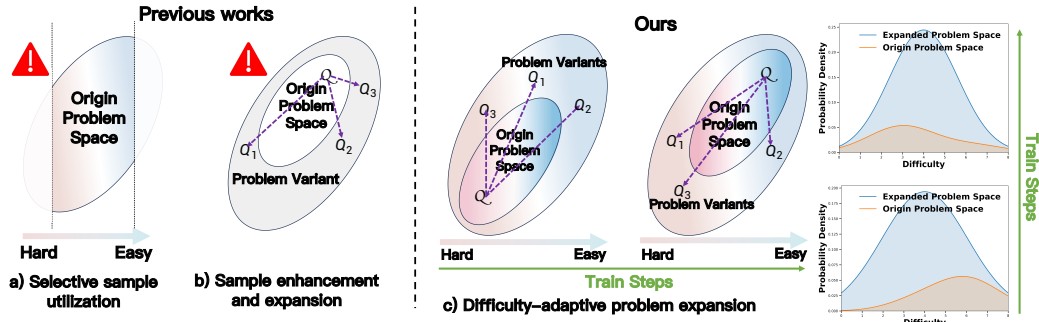

Figure 1: (a) Selective sample utilization relies on only a subset of data, leading to underuse. (b) Sample enhancement expands data without difficulty awareness, causing even severe advantage sparsity. (c) Our method adaptively expands the sample space by problem difficulty, ensuring a stable difficulty distribution.

2024) and multimodal logical reasoning (Huang & Chang, 2022; Guo et al., 2025). Nevertheless, the heterogeneous nature of text and visual modalities makes long-chain reasoning challenging, requiring both careful observation and stepwise problem-solving (Xu et al., 2024). To address these challenges, recent studies have explored multimodal chain-of-thought approaches (Zhou et al., 2024; Chen et al., 2024a; Wang et al., 2025b), including Video-of-Thought (Fei et al., 2024), Det-CoT (Wu et al., 2024a), CoI (Meng et al., 2023), and Grounded-RL (Sarch et al., 2025), which aim to improve reasoning by decomposing problems or leveraging structured strategies. Complementary to these advances, reinforcement learning (RL) has emerged as a powerful framework for further enhancing MLLMs: proximal policy optimization (PPO) (Schulman et al., 2017) and direct preference optimization (DPO) (Rafailov et al., 2023) are widely used for alignment, while GRPO (Shao et al., 2024) advances long-chain reasoning by combining rewards with group-relative advantage estimation, significantly boosting both alignment and reasoning efficiency.

However, directly applying GRPO to MLLMs faces a critical challenge—advantage vanishing (Huang et al., 2025; Park et al., 2025; Yao et al., 2025; Wang et al., 2025a; Meng et al., 2025; Zhang et al., 2025; Zhang & Zuo, 2025). GRPO computes relative advantages by comparing sampled responses for a given problem and then standardizes the advantages across all samples. When a problem is too easy (or too difficult) for the current model, obtaining too few positive (or negative) rewards can lead to excessively large advantage values, causing high gradient noise and unstable training. Under extreme conditions, all responses may be entirely correct or incorrect, yielding zero relative advantage. This slows learning and weakens the model's reasoning performance across varying difficulties. Existing approaches attempt to address this issue in several ways. One approach is sample enhancement and expansion, which enlarges the problem space by adding prompts to difficult instances or generating diverse text and image variants, though it may exacerbate advantage vanishing due to poor control over the difficulty distribution (Huang et al., 2025; Park et al., 2025; Yao et al., 2025)and introduce hidden biases (Gao et al., 2025). Another approach is selective sample utilization, which prioritizes highly effective samples or disregards those with low learning contribution to improve efficiency; however, this may limit exposure to complex problems or reduce data diversity (Wang et al., 2025a; Meng et al., 2025). Finally, indirect reward design provides finer-grained feedback signals to mitigate reward sparsity, but these rewards may not perfectly align with ultimate task objectives, potentially guiding the model toward suboptimal directions (Zhang et al., 2025; Zhang & Zuo, 2025).

Although the above methods can mitigate certain instances of advantage vanishing, they all overlook a critical issue: *as shown in Figure 1(c), as training progresses, problem difficulty continuously decreases, advantage vanishing intensifies, and model training efficiency steadily declines.* To address this, we propose DIVA-GRPO, an adaptive method that dynamically adjusts problem difficulty and variant distribution while preserving semantic consistency, and integrates both local (individual problem) and global (multiple variants derived from the original problem) reward computation. During training, problem difficulty is iteratively assessed based on model responses: simple problems are augmented with complex text and image perturbations, moderately difficult ones with diverse text variants, and hard ones with "think-step" reasoning variants, ensuring effective sample utilization and mitigating advantage vanishing. To balance feedback across different difficulty levels, DIVA-GRPO applies batch z-score normalization and difficulty-weighted scaling, and further introduces reward-range-based rescaling to prevent inflated advantages from minor reward differences. This

enables stable policy optimization, encourages exploration, and enhances long-chain reasoning, while remaining broadly applicable to GRPO-based methods for improved optimization efficiency.

Extensive experiments validate the effectiveness and efficiency of DIVA-GRPO. Our model excels across six challenging benchmarks, achieving state-of-the-art performance at the 7B scale, approaching the accuracy of much larger models and proprietary commercial systems. Moreover, DIVA-GRPO substantially accelerates training. Using the same set of original data, it reaches the optimal performance with over a $2.55\times$ reduction in required steps and delivers more than a $1.76\times$ end-to-end speedup in wall-clock time, all while preserving stable and informative advantage signals across difficulty levels. These results highlight the substantial improvements in both performance and training efficiency offered by our model.

Overall, DIVA-GRPO overcomes the limitations of conventional GRPO by combining adaptive difficulty assessment, variant generation, and robust advantage estimation. This framework alleviates reward sparsity and advantage vanishing, while enhancing reasoning stability and performance across a wide range of multimodal tasks, from visual question answering to complex logical reasoning. Our core contributions are as follows:

- Propose DIVA-GRPO, based on dynamic difficulty assessment, using difficulty-adaptive variant generation and advantage sharing to alleviate reward sparsity and advantage vanishing.
- Introduce joint estimation of local and global advantages, combined with batch z-score normalization and difficulty-weighted scaling, to improve training stability.
- Present a reward-range-based advantage rescaling method, effectively preventing unreasonable advantage inflation and accelerating convergence.
- Conduct experiments on six mainstream multimodal reasoning benchmarks, demonstrating the effectiveness and superiority of DIVA-GRPO.

## 2 CHALLENGE: REWARD SPARSITY AND ADVANTAGE VANISHING

We begin by reviewing the key concepts of Group Relative Policy Optimization (GRPO), which form the foundation of our difficulty-adaptive variant strategy.

**Group Relative Policy Optimization (GRPO).** Let $\mathcal{Q} = \{q_1, q_2, \ldots, q_N\}$ denote the set of training problems. For a given problem $q$, the model $\pi_\theta$ generates $k$ candidate responses $\mathcal{Y}_q = \{y_1, y_2, \ldots, y_k\}$ through rollouts. Each response $y_i$ receives a scalar reward $r(y_i) \in \mathbb{R}$ determined by a reward function, which can be either a rule-based evaluator (e.g., answer correctness) or a learned reward model. GRPO computes the *relative advantage* of each response within its group by standardizing its reward:

$$A(y_i) = \frac{r(y_i) - \mu_r}{\sigma_r + \epsilon}, \quad \mu_r = \frac{1}{k}\sum_{j=1}^{k} r(y_j), \quad \sigma_r = \sqrt{\frac{1}{k}\sum_{j=1}^{k}(r(y_j) - \mu_r)^2},$$

where $\epsilon$ is a small constant for numerical stability. The policy gradient is then estimated as

$$\nabla_\theta \mathcal{L}(\theta) = \mathbb{E}_{q \sim \mathcal{Q},\, y \sim \pi_\theta}\big[A(y)\,\nabla_\theta \log \pi_\theta(y|q)\big].$$

**Reward Sparsity and Advantage Vanishing.** GRPO and its extensions often face **reward sparsity**: when the base multimodal model has limited capability or the problem is difficult, only a few reasoning paths obtain positive rewards, especially in early training. They also suffer from **advantage vanishing**: since relative advantages are computed within response groups, overly hard or easy problems lead to all-correct or all-wrong outputs, yielding zero advantages. As training proceeds, this issue worsens, reducing optimization efficiency and sample utilization.

**Motivation.** Existing solutions address these problems through (i) *sample enhancement and expansion*, which enlarges the problem space without adjusting intrinsic difficulty; (ii) *sample selection and utilization*, which improves efficiency but risks discarding valuable hard cases and reducing diversity; and (iii) *indirect reward design*, which enriches supervision but may bias optimization. While partially effective, these methods cannot guarantee stable reward variance within each group.

This motivates our central question: ***how can we preserve stable and informative reward variance regardless of problem difficulty?*** We argue that the key lies in dynamically assessing problem

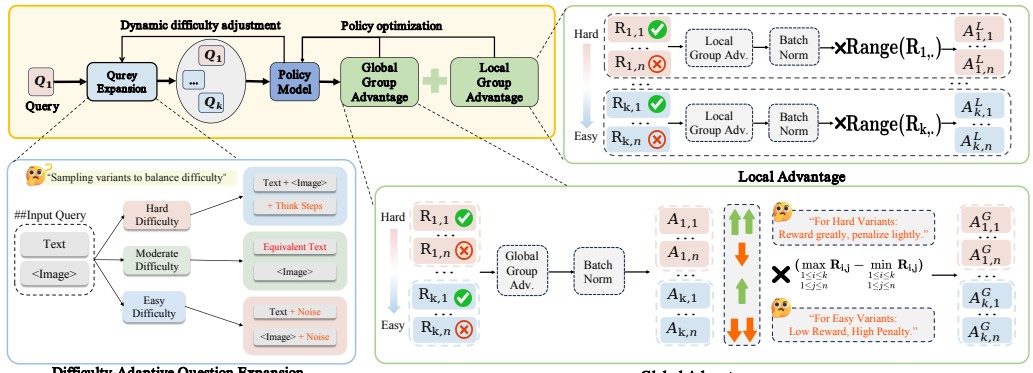

Figure 2: Overview of the proposed **DIVA-GRPO** method. For a given question, we dynamically assess its difficulty based on past rollout rewards and adaptively sample variants of different difficulty levels. As shown, when the original question is hard, easier variants are sampled to ensure reward diversity. We then compute local (the question itself) and global (the question with its variants) advantages, and obtain the final advantage through difficulty-aware reweighting and reward-range rescaling to update the policy model.

difficulty and adaptively adjusting the sampling of variants, ensuring every problem produces both positive and negative feedback.

## 3 DIFFICULTY-ADAPTIVE VARIANT ADVANTAGE WITH GRPO

To address this challenge, we design a difficulty-adaptive method, **DIVA-GRPO**. Its core idea is to dynamically assess problem difficulty, adaptively sample semantically consistent variants of different difficulties, and enhance the diversity of rewards within the problem and its variant space. By computing difficulty-weighted local (for the original problem) and global (for the problem and its variants) group advantages, the framework mitigates reward sparsity and vanishing advantage issues. Figure 2 illustrates the overall pipeline.

### 3.1 DIFFICULTY ASSESSMENT OF SAMPLES

A prerequisite for implementing adaptive strategies is to assign an appropriate difficulty level to each problem. Notably, problem difficulty is neither inherent nor static; rather, it should be dynamically assessed in line with the evolving capability of the model. To this end, DIVA-GRPO introduces a difficulty assessment mechanism based on historical rollout outcomes: if most rollouts for a problem are correct, it is regarded as relatively easy for the current model; otherwise, it is considered relatively difficult. This assessment is recalibrated at every training epoch, ensuring that perceived difficulty adapts as the model improves. The resulting difficulty estimates are then used to guide variant generation and sampling strategies, enabling the model to balance exploration and exploitation while maintaining meaningful advantage signals continuously.

We define problem difficulty within the range $[D_{\min}, D_{\max}]$, where $D_{\min}$ represents the easiest level and $D_{\max}$ the most difficult. At the beginning of training, all problems are initialized at the midpoint of this range: $D_{\mathrm{mid}} = \frac{D_{\min} + D_{\max}}{2}$. Let each original problem contain $m$ variants, and let each variant be rolled out $k$ times. We denote the empirical accuracy of these rollouts as $\alpha = \frac{1}{mk} \sum_{i=1}^{m} \sum_{j=1}^{k} \mathbb{I}[y_{i,j} \text{ is correct}]$, where $\mathbb{I}[\cdot]$ is the indicator function. The difficulty is then updated according to the following rule:

$$D^{\mathrm{new}} = \mathrm{clip}\left(D^{\mathrm{old}} + \eta \cdot (0.5 - \alpha),\ D_{\min},\ D_{\max}\right),$$

where $\eta > 0$ is a learning rate controlling the adjustment magnitude, and $\mathrm{clip}(\cdot, D_{\min}, D_{\max})$ ensures that the difficulty remains within $[D_{\min}, D_{\max}]$. This update rule guarantees that:

- if most rollouts are correct ($\alpha \to 1$), the difficulty decreases;
- if most rollouts fail ($\alpha \to 0$), the difficulty increases;
- if the accuracy is around $50\%$, the difficulty remains stable.

In this way, difficulty levels evolve smoothly with model performance, allowing more nuanced adjustments than the simple "all-correct/all-wrong" rule, and ensuring continuous calibration of sampling strategies throughout training. For the hyperparameter $\eta$, ablation experiments and recommendations for its selection can be found in Appendix D.2.

## 3.2 DIFFICULTY-ADAPTIVE VARIANT GENERATION

We aim to provide each problem with an appropriate advantage. To this end, we introduce a method for generating adaptive variants based on the difficulty assessment of each problem derived above. Let each original problem be denoted as $q = (I_q, T_q)$, where $I_q$ is the associated image and $T_q$ is the textual description and question. After evaluating the difficulty $D_q \in [D_{\min}, D_{\max}]$ of each problem, we generate a set of difficulty-specific consistent variants $\mathcal{V}_q = \{q^{(1)}, q^{(2)}, \dots\}$ for training, ensuring that each variant preserves the original answer $y_q^*$ while selectively adjusting its difficulty to optimize the model's learning. By dynamically adjusting the characteristics of each variant according to $D_q$, we aim to ensure that the reward distribution for each problem provides meaningful advantage signals while maintaining a more balanced profile. The sampling method is described in Appendix D.2, and several case studies of the variants are presented in Appendix H.

- **Simpler problems** ($D_q < D_{\mathbf{mid}}$): Variants are generated by perturbing both the text and the image: $q^{(i)} = (I_q^{(i)}, T_q^{(i)})$. Textual variants $T_q^{(i)}$ are created by rephrasing, restructuring, or introducing slight linguistic perturbations to the original question, ensuring the answer remains unchanged. These modifications enhance the model's sensitivity to subtle differences, thereby improving robustness. Image variants $I_q^{(i)}$ are generated through perturbation functions such as rotation, Gaussian noise, salt-and-pepper noise, speckle noise, and blurring. Stronger or multiple perturbations increase difficulty while preserving correctness. Alternatively, the textual information $T_q$ can be embedded within the images in the variant set $\mathcal{V}_q$ and provided as images to convey the problem information.

- **Moderate problems** ($D_q \approx D_{\mathbf{mid}}$): Variants are generated by creating semantically equivalent textual versions: $\mathcal{V}_q = \{(I_q, T_q^{(i)}) \mid T_q^{(i)} \text{ is a paraphrase of } T_q\}$. These variants maintain the original difficulty while diversifying expression. This allows the model to experience multiple formulations of the same problem, improving generalization to unseen expressions.

- **Difficult problems** ($D_q > D_{\mathbf{mid}}$): Variants incorporate partial reasoning guidance: $q^{(i)} = (I_q, T_q \oplus R_q^{(i)})$, where $R_q^{(i)}$ is a sequence of intermediate reasoning steps generated and verified by a closed-source model. For more difficult problems, additional reasoning steps are provided as hints. This ensures that even challenging problems produce meaningful advantage signals, mitigating gradient vanishing and promoting gradual mastery of complex reasoning.

## 3.3 DIFFICULTY-WEIGHTED AND NORMALIZED ADVANTAGE BALANCING

Before introducing our balancing strategy, we first review the notion of local and global advantages used in GRPO-based training. While GRPO computes advantages within a single problem, recent work Yao et al. (2025) introduces semantically consistent variants to enable broader comparisons. Let an original problem be denoted as $q = (I_q, T_q)$, with associated image $I_q$ and textual description $T_q$. For each $q$, a set of variants $\mathcal{V}_q = \{q^{(1)}, \dots, q^{(N)}\}$ is constructed, modifying only text or image while preserving the ground-truth answer. Two types of advantages are defined:

- **Local advantage:** $A_{\text{local}}(y_i^{(i)})$, computed for each problem using the standard GRPO formula.
- **Global advantage:** $A_{\text{global}}(y_i^{(j)})$, computed across all responses in the variant set: $A_{\text{global}}(y_i^{(j)}) = \frac{r(y_i^{(j)}) - \mu_q}{\sigma_q + \epsilon}$, where $\mu_q$ and $\sigma_q$ are the mean and standard deviation of rewards across all responses in $\mathcal{V}_q$.

To ensure that both local and global advantages contribute effectively while accounting for varying problem difficulty, we propose a two-step balancing strategy: (1) *normalization* to make local and global signals comparable, and (2) *difficulty-weighted scaling* to adaptively rescale advantages according to each problem's dynamic difficulty coefficient. This design stabilizes optimization by preventing dominance of global advantages and aligning reward signals with problem difficulty.

Formally, after sampling multiple variants for each problem, we compute both local and global advantages. However, two issues arise in this process.

**(1) Local–Global imbalance.** The magnitudes of local and global advantages are unequal: local advantages are computed from $k$ rollouts, whereas global advantages are based on $m \times k$ samples. Consequently, global advantages tend to be larger and dominate the optimization, while local signals are underweighted. A detailed analysis of the distributions of the original local and global advantages can be found in Appendix D.6. To address this issue, we apply batch-level $z$-score normalization separately:

$$\tilde{A}_{\text{local}}(y) = \frac{A_{\text{local}}(y) - \mu_{\text{local}}}{\sigma_{\text{local}} + \epsilon}, \quad \tilde{A}_{\text{global}}(y) = \frac{A_{\text{global}}(y) - \mu_{\text{global}}}{\sigma_{\text{global}} + \epsilon},$$

where $\mu_{\text{local}}, \sigma_{\text{local}}$ and $\mu_{\text{global}}, \sigma_{\text{global}}$ are the mean and std of local and global advantages within a batch. This ensures that both signals remain comparable and contribute fully to training.

**(2) Difficulty-weighted scaling.** Existing methods treat easy and difficult problems equally when computing advantages, without accounting for their varying difficulty levels. To encourage the model to tackle harder problems, we introduce difficulty-weighted scaling after normalization.

Let $\{D_q^{(i)}\}_{i=1}^N$ denote the difficulty coefficients of the $N$ variants in a problem group $\mathcal{V}_q$, and let $\bar{D}_q = \frac{1}{N} \sum_{i=1}^N D_q^{(i)}$ be the group-wise mean difficulty. For each response $y_i$ associated with variant $q^{(i)}$, the rescaled advantage is computed as

$$\hat{A}(y_i \mid q^{(i)}) = \exp\left( k \cdot (D_q^{(i)} - \bar{D}_q) \cdot \text{sgn}(\tilde{A}(y_i)) \right) \cdot \tilde{A}(y_i),$$

where $\tilde{A}(y_i)$ is the normalized advantage, $\text{sgn}(\cdot)$ is the sign function, and $k > 0$ is a sensitivity. The reason for using relative difficulty weighting is analyzed and supported by ablation experiments in Appendix D.5, while the analysis of the sensitivity parameter $k$ are presented in Appendix D.1.

Intuitively, when a variant is harder than the group average ($D_q^{(i)} > \bar{D}_q$), correct answers ($\tilde{A} > 0$) are amplified while incorrect ones are softened. Conversely, when a variant is easier than average, correct answers are down-weighted and incorrect ones are penalized more heavily.

In this way, the training process achieves difficulty-adaptive optimization, effectively balancing the contributions of both the relative difficulty within a problem group and the magnitudes of advantages. The theoretical validity of this balancing strategy is rigorously established in Appendix B, where Theorem B.1 demonstrates that reducing gradient variance accelerates convergence, and Corollary B.2 shows that our normalization and weighting strategy significantly decreases variance while preserving unbiased gradient estimates. Moreover, Appendix C mathematically confirms that the optimization signal is strongest when the ratio of correct to incorrect samples is approximately 1:1, providing a solid theoretical foundation for our dynamic difficulty adjustment mechanism.

### 3.4 Reward-Range-Based Advantage Rescaling (RRB-Rescaling)

In GRPO-based reinforcement learning, we observed that advantage estimation can become unreliable when the reward range within a rollout group is small. If rewards are tightly clustered, standard $z$-score normalization may exaggerate minor differences, producing misleading optimization signals. As a result, the model may over-reward trivial gains while overlooking substantial ones.

Consider a reward scheme where correctly formatted responses receive $0.1$ and fully correct responses receive $0.9$. Suppose we have two samples, each rolled out 5 times: Sample A with rewards $[0, 0, 0, 0, 0.1]$ and Sample B with rewards $[0, 0, 0, 0, 1]$. After applying standard $z$-score normalization, both groups assign the same advantage ($-0.45$) to the zero-reward rollouts and the same high advantage ($1.79$) to the last rollout. This overestimates the gain from minor formatting correctness in Sample A and underestimates the greater achievement in Sample B.

To address this issue, we propose *reward-range-based advantage rescaling*. Let $\mathcal{R}_q = \{r_1, r_2, \ldots, r_k\}$ denote the rewards of all rollouts for a problem $q$, with the maximum possible reward range $R_{\max}$. We define the reward range and the rescaled advantage as

$$\Delta r_q = (\max(\mathcal{R}_q) - \min(\mathcal{R}_q))/R_{\max}, \quad \hat{A}_{\text{range}}(y_i) = \Delta r_q \cdot \tilde{A}(y_i)$$

where $\tilde{A}(y_i)$ is the normalized advantage obtained via standard z-score. Intuitively, this method ensures that the magnitude of the advantage reflects the actual variability in rewards, preventing

Table 1: Performance comparison across multimodal mathematical benchmarks. Bold denotes the best performance among 7B models, and underline marks the best overall performance. Evaluation is conducted with VLMEvalKit (Duan et al., 2024), while results for other models are taken from Meng et al. (2025) and Yao et al. (2025). For each entry, the score before "/" is our re-evaluation using the officially released checkpoints, and the score after "/" is reported in the original paper.

| Model | MathVista | MathVerse | MathVision | OlympiadBench | WeMath | MMK12test | Avg. |
|---|---|---|---|---|---|---|---|
| Claude3.7-Sonnet | 66.8 | 52.0 | 41.3 | *48.9* | 72.6 | 55.3 | 56.15 |
| GPT-4o | 63.8 | 50.2 | 30.4 | *35.0* | 68.8 | 49.9 | 49.68 |
| o1 | 73.9 | 57.0 | 60.3 | 68.0 | 98.7 | 73.9 | 72.30 |
| Gemini2-flash | 70.4 | 59.3 | 41.3 | 51.0 | 71.4 | 65.2 | 59.77 |
| InternVL2.5-VL-8B | 64.4 | 39.5 | 19.7 | 12.3 | 53.5 | 45.6 | 39.17 |
| Qwen-2.5-VL-7B | 68.2 | 47.9 | 25.4 | 20.2 | 62.1 | 53.6 | 46.23 |
| InternVL2.5-VL-38B | 71.9 | 49.4 | 31.8 | 32.0 | 67.5 | 58.0 | 51.77 |
| Qwen-2.5-VL-32B | 71.7/74.7 | 49.9 | 40.1 | 30.0 | 69.1 | 66.8 | 54.6 |
| InternVL2.5-VL-78B | 72.3 | 51.7 | 32.2 | 31.1 | 66.3 | 61.6 | 52.53 |
| Qwen-2.5-VL-72B | 74.8 | 57.6 | 38.1 | 40.4 | 72.4 | 70.5 | 59.0 |
| InternVL2.5-8B-MPO | 68.9 | 35.5 | 21.5 | 7.8 | 53.5 | 34.5 | 36.95 |
| InternVL2.5-38B-MPO | 73.8 | 46.5 | 32.3 | 25.6 | 66.2 | 48.3 | 48.78 |
| QVQ-72B-Preview | 71.4 | 48.2 | 35.9 | 33.2 | 65.4 | 61.5 | 52.60 |
| Adora-7B | 73.5 | 50.1 | 23.0 | 20.1 | 64.2 | 58.1 | 48.17 |
| R1-Onevision-7B | 64.1 | 47.1 | 23.5/29.9 | 17.3 | 61.8 | 39.8 | 42.27 |
| OpenVLThinker-7B | 70.2 | 47.9 | 25.3 | 20.1 | 64.3 | 60.6 | 48.07 |
| MM-Eureka-7B | 71.7/73.0 | 50.3 | 26.9 | 20.1 | 66.1 | 64.5 | 49.93 |
| R1-ShareVL-7B | 73.5/75.4 | 52.8 | 29.5 | 21.3 | 67.9 | 68.8 | 52.30 |
| SFT-7B | 66.4 | 49.6 | 21.3 | 16.2 | 57.6 | 54.3 | 44.23 |
| DIVA-GRPO-7B (Ours) | **74.2** | **57.6** | **32.1** | **23.1** | **69.3** | **70.2** | 54.58 |

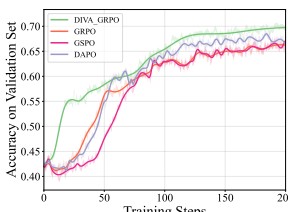

(a) RQ1: Effectiveness of the DIVA-GRPO

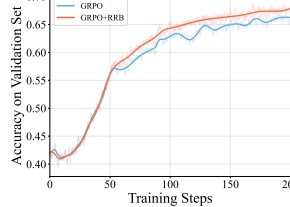

(b) RQ3: Effectiveness of RRB on General GRPO Methods

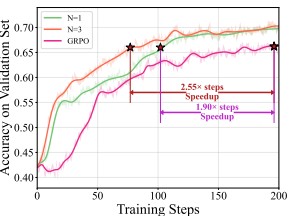

(c) RQ4: Impact of DIVA-GRPO on Efficiency and Speed

Figure 3: Effect of Training Steps on Model Performance on the Validation Set.

minor differences from being exaggerated and providing a more reliable optimization signal. This rescaling method can be applied independently of difficulty-weighted scaling and is compatible with any GRPO-based framework, improving the stability and efficiency of policy optimization.

## 4 EXPERIMENTS

In this section, we conduct extensive experiments to evaluate the effectiveness of DIVA-GRPO, complemented by ablation studies to assess the contributions of its key components. Our evaluation is driven by the following research questions:

- **RQ1 (Effectiveness)**: How does DIVA-GRPO compare to recent advanced systems?
- **RQ2 (Completeness)**: What impact does removing key components have on model performance?
- **RQ3 (Generalization)**: How well does RRB-Rescaling generalize to standard GRPO?
- **RQ4 (Efficiency)**: What impact does DIVA-GRPO have on training efficiency?

### 4.1 EXPERIMENTAL SETUP

**Benchmarks.** Our evaluation covers six diverse benchmarks: *MathVista*(Lu et al., 2023), *MathVerse*(Zhang et al., 2024), *MathVision*(Wang et al., 2024), *OlympiadBench*(He et al., 2024), *WeMath*(Qiao et al., 2024), and *MMK12-test*(Meng et al., 2025).

**Baselines.** Our comparisons cover three categories of models: *(i) closed-source proprietary MLLMs*; *(ii) open-source base MLLMs*; *(iii) open-source fine-tuned MLLMs*.

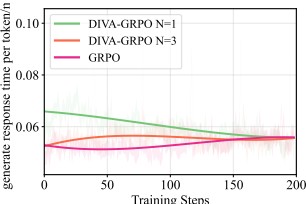 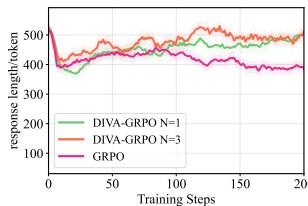 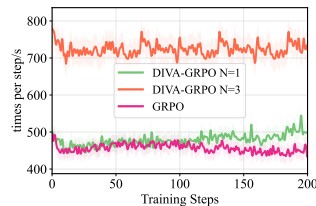

(a) Time per token generation      (b) Average length of response      (c) Time per step

Figure 4: Statistics of time and training speed during training and testing, showing that our method only leads to a slight increase in training time per step, primarily due to the increased output length.

**Implementation.** Our method is trained on **Qwen2.5-VL-7B-Instruct** (Bai et al., 2025) with AdamW at a learning rate of $10^{-6}$. We employ the latest EasyR1 framework for training. Specifically, for methods that do not generate variants, we set $k = 10$, while for methods that do generate variants, we set $k = 5$, using both $N = 1$ and $N = 3$ settings to ensure the fairness of the experiment. This setup guarantees a rigorous and fair comparison across all baselines. Difficulty scores $D_q$ are initialized to $5$ ($D_{\min} = 1$, $D_{\max} = 9$) with $\eta = 4$. The detailed parameter analysis of can be found in Appendix D. Textual reasoning hints are pre-generated offline, while image perturbations are applied online. All textual variants and reasoning sequences are generated offline using GPT-o3.

## 4.2 Main Results (**RQ1: Effectiveness**)

We report results on 7B-scale models, with the main experiments conducted on the R1-ShareVL-52K dataset. DIVA-GRPO demonstrates superior performance compared to both the base models and other multimodal training approaches. As shown in Table 1, we provide a comprehensive comparison against state-of-the-art systems across five widely used and challenging benchmarks.

Our method achieves state-of-the-art performance across all datasets at the **7B scale**, delivering consistently strong results on both Chinese and English benchmarks with an average score of 54.58. Remarkably, on *MathVista*, *MathVerse*, and *WeMath*, its performance is already on par with the much larger Qwen2.5-VL-72B, while substantially outperforming several open-source base models (e.g., Qwen2.5-VL-32B, InternVL2.5-VL-78B) and proprietary systems (e.g., GPT-4o, Claude 3.7-Sonnet), showcasing notable efficiency and cost-effectiveness. Nevertheless, on more challenging competition-level mathematics tasks, our method still falls short of ultra-large open-source models and cutting-edge proprietary systems, likely due to inherent limitations in model capacity and training data coverage. We further explored applying direct SFT to the backbone model using reasoning traces generated by GPT-o3 and found that such direct training not only failed to improve performance but even underperformed the base model—further underscoring the effectiveness of our approach. Compared with its backbone model, Qwen2.5-VL-7B, our method consistently improves results across all benchmarks, achieving an average accuracy gain of 8.23 points, which highlights its clear advantages over existing multimodal training frameworks. The discussion of different external models can be found in Appendix D.4.

Figure 5 further illustrates the training dynamics: in the early stage, samples have not yet formed a clear difficulty distribution, with most concentrated around medium difficulty and exhibiting only small advantages. As training progresses, the distribution gradually expands toward both easier and harder problems, and advantage signals become more distinct across difficulty levels. In later stages, the model maintains informative and balanced advantage signals even on high-difficulty samples, rather than collapsing into trivial "all-correct" or "all-wrong" states. These dynamics highlight the effectiveness of DIVA-GRPO in sustaining stable and efficient optimization.

## 4.3 Ablation Study

**RQ2 (Completeness)**: Component Ablation of DIVA-GRPO

We perform ablation studies on the three core components of DIVA-GRPO at the 7B scale using representative benchmarks (*MathVista*, *MathVerse*, and *MMK12test*). Experiments are conducted on 5,000 randomly sampled instances from the *MMK12* dataset to ensure timely and fair comparisons. Each 20 steps constitutes one epoch, with a total of 10 epochs of training. We evaluate the impact of

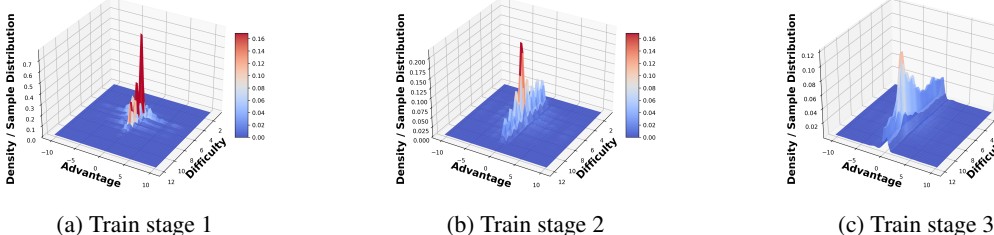

| (a) Train stage 1 | (b) Train stage 2 | (c) Train stage 3 |

Figure 5: 3D kernel density estimation (KDE) surfaces of the joint distribution between problem difficulty and global advantage across different training stages. The surface height reflects the sample density, illustrating how the model's learning dynamics evolve with respect to difficulty and advantage.

removing Adaptive Variant Generation (including Global-Local Advantage and Balance), Difficulty Weighting, Reward-Range-Based Advantage Rescaling (RRB-Rescaling), and Global-Local Balance (G-L Balance), and compare each variant with the full model.

As reported in Table 2, removing any single component consistently decreases performance, with the full DIVA-GRPO model achieving the highest accuracy across all benchmarks. These results indicate that all components contribute complementary gains and none can be omitted without performance degradation, highlighting the necessity of the complete model design.

Furthermore, as shown in 3a, we compare DIVA-GRPO with GRPO, GSPO Zheng et al. (2025), and DAPO Yu et al. (2025) throughout the entire training process. Experimental results demonstrate that, under strict control of time and total response size, our method outperforms GRPO, GSPO, and DAPO in terms of both training speed and stability. Additionally, we have evaluated the performance of our method on Qwen3-VL-4B and 8B, with the detailed results presented in Appendix D.3.

Table 2: Ablation study of DIVA-GRPO, showing that each component provides gains and the full model achieves the best performance (accuracy, %).

| Method | MathVista | MathVerse | MMK12test | Avg. |
|---|---|---|---|---|
| w/o Variant Generation | 70.0 | 53.7 | 61.1 | 61.6 |
| w/o Difficulty-Weighting | 69.9 | 55.7 | 66.5 | 64.0 |
| w/o RRB-Rescaling | 71.5 | 55.2 | 64.7 | 63.8 |
| w/o G-L Balance | 70.8 | 55.4 | 66.0 | 64.1 |
| **Full DIVA-GRPO** | **73.2** | **56.3** | **68.8** | **66.1** |

**RQ3 (Generalization)**: Generalization of RRB-Rescaling

To verify the broader applicability of Reward-Range-Based Advantage Rescaling (RRB-Rescaling), we incorporate it into standard GRPO without any other modifications. Table 3 shows that adding RRB-Rescaling increases the average accuracy from 60.23 to 62.17, with improvements observed across all three benchmarks. This is further illustrated in Figure 3b. This demonstrates that RRB-Rescaling enhances both stability and performance in a general GRPO setting, confirming that this component is not limited to DIVA-GRPO but can benefit other GRPO-style training frameworks.

Table 3: Evaluation of RRB-Rescaling on standard GRPO.

| Method | MathVista | MathVerse | MMK12test | Avg. |
|---|---|---|---|---|
| GRPO (base) | 69.3 | 52.8 | 58.6 | 60.23 |
| + RRB | **70.0** | **53.6** | **62.9** | **62.17** |

**RQ4 (Efficiency)**: Training Efficiency and Stability

We compare the training performance of DIVA-GRPO and GRPO on Qwen2.5-VL-7B. As shown in Figure 3c, we measure the number of steps required by DIVA-GRPO to reach the test-set optimum within 10 epochs of GRPO training. When GRPO uses $n = 10$ and DIVA-GRPO uses $n = 5$, we ensure a constant number of sampled sentences when $N = 1$ and a consistent amount of input data when $N = 3$. Figure 4a shows the time consumed per token during generation, Figure 4b illustrates the average response length, and Figure 4c depicts the time per step. Next, we calculate the speedup of DIVA-GRPO in terms of both steps and time.

- **For $N = 1$**, we ensure the consistency of the response count, meaning the total number of samples remains equal, while DIVA-GRPO accelerates the steps by a factor of $1.90\times$. According to the statistics, while the time per token increases by only 13% and the time per step by just 4%, the total time comparison shows that our method consumes an average of 7.98 minutes per step, compared to 7.61 minutes per step for the GRPO method. For generating variants and think steps,

approximately 32 minutes are required for every 5000 samples, while the generation of visual variants is almost instantaneous and does not incur additional time costs, as it is already included in the training time. Even when factoring in the time for generating variants, our method achieves a $1.76\times$ speedup in terms of time.

- **For** $N = 3$, we ensure that the data requirements remain unchanged, while DIVA-GRPO accelerates the steps by a factor of $2.55\times$. In this case, the average time per step is 12.03 minutes. Even when accounting for the time required to generate variants, our method achieves a $1.56\times$ speedup in terms of time. This clearly demonstrates the efficiency of our approach.

## 5 RELATED WORK

With the increasing application of GRPO in MLLMs to enhance reasoning, several challenges remain unresolved, particularly sparse rewards and advantage vanishing. Existing solutions can be grouped into three main approaches: sample augmentation and problem-space expansion, sample selection and utilization, and reward design.

**Sample augmentation and expansion** methods aim to enlarge the problem space using difficulty-aware mechanisms or diverse sample generation to improve exploration and generalization. Noisy Rollout (Liu et al., 2025a) uses image noising to facilitate reinforcement learning, introducing visual diversity to prevent overfitting and improve robustness. Hint-GRPO (Huang et al., 2025) adapts hints based on task difficulty, enhancing data efficiency for hard samples. DeepVideo-R1 (Park et al., 2025) adjusts difficulty to help variants regain advantage, though it does not reshape single-sample difficulty distributions, leaving sparse rewards in extreme cases. R1-shareVL (Yao et al., 2025) expands the space with multiple variants and applies global advantage to shift difficulty, but lacks a complete mechanism to mitigate sparsity.

**Sample selection and utilization** methods focus computation on effective data while ignoring unhelpful samples, improving efficiency but risking premature abandonment of hard cases. VL-Rethinker (Wang et al., 2025a) uses selective replay to alleviate reward sparsity and preserves advantage diversity by replaying strong past samples. MM-Eureka (Meng et al., 2025) selects prompts with partial correctness, filtering zero-advantage cases to stabilize training. While helpful, these methods do not maximize overall sample utilization or provide a global optimization view.

**Reward design** methods aim to provide denser, continuous signals to reduce sparsity and stabilize reasoning. MM-Eureka introduces a hyperparameter $\lambda$ to weight format rewards. R1-VL (Zhang et al., 2025) proposes StepGRPO, incorporating reasoning-accuracy and step-wise validity rewards for denser feedback. The introduced indirect rewards may therefore deviate from the primary objective and potentially mislead the model. DeepVideo-R1 introduces Reg-GRPO, transforming GRPO into a regression loss, while GRPO-LEAD (Zhang & Zuo, 2025) weights hard problems in advantage computation via dynamic difficulty awareness. By removing the variance term in advantage estimation, Dr. GRPO (Liu et al., 2025b) seeks to reduce variance and stabilize the learning signal. This shares the motivation with our RRB method, as both aim to provide more stable learning signals. Although these methods mitigate convergence and stability issues, they fail to address the fundamental challenges of sparse rewards and diminishing advantages.

## 6 CONCLUSION

In this work, we identified and addressed a fundamental limitation of GRPO-based reinforcement learning for multimodal large language models: reward sparsity and vanishing advantages, which hinder effective long-chain reasoning. To overcome this, we proposed DIVA-GRPO, a difficulty-adaptive variant advantage framework that dynamically assesses problem difficulty, generates tailored variants, and computes both local and global advantages with difficulty-aware normalization and reward-range-based rescaling. This design ensures stable and informative optimization signals across problems of varying difficulty, mitigates reward sparsity, and enhances training stability. Extensive experiments on six multimodal reasoning benchmarks demonstrated that DIVA-GRPO consistently outperforms existing GRPO-based methods in reasoning accuracy, convergence speed, and training efficiency. Overall, our approach provides a generalizable framework for improving reinforcement learning in MLLMs, offering a principled way to balance exploration, sample utilization, and advantage estimation for complex multimodal reasoning tasks.

## ACKNOWLEDGMENTS

This work was supported by the Strategic Priority Research Program of the CAS under Grants No.XDB0680302, the National Natural Science Foundation of China (NSFC) under Grants No. 62276248, the Key Research and Development Program of Xinjiang Uyghur Autonomous Region Grant No. 2024B03026, the Beijing Nova Program under Grants No. 20250484765, and the Youth Innovation Promotion Association CAS under Grants No. 2023111.

## ETHICS STATEMENT

In this study, all data collection and analysis were conducted in compliance with ethical standards, and no human subjects or animal experiments were involved. The research did not use any sensitive personal data or privacy information, and all experimental designs and data processing methods adhered to relevant legal and ethical guidelines. Regarding dataset releases, we have strictly followed applicable academic ethics regulations, ensuring that all data and code sharing comply with legal and ethical requirements. There are no conflicts of interest in this study, and all funding sources and support have been clearly disclosed. The study ensures transparency, integrity, and fairness, strictly adhering to relevant research ethics and compliance standards.

## REPRODUCIBILITY STATEMENT

To ensure full reproducibility of our experiments, we provide all code, configurations, and instructions required to replicate our results in the repository: https://github.com/Siaaaaaa1/DIVA-GRPO. The repository includes scripts for training and evaluation, pre-processing routines, and detailed README documentation describing the dependencies and steps to reproduce the reported results.

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

## A    DETAILED BENCHMARKS AND BASELINES

### A.1    BENCHMARKS

We evaluate six multimodal mathematical reasoning benchmarks:

- **MathVista**: *A comprehensive benchmark combining challenges from diverse mathematical and visual tasks. It consists of 6,141 examples, derived from 28 existing multimodal datasets involving mathematics and 3 newly created datasets (i.e., IQTest, FunctionQA, and PaperQA).*

- **MathVerse**: *An all-around visual math benchmark designed for an equitable and in-depth evaluation of Multi-modal Large Language Models (MLLMs). It includes 2,612 high-quality, multi-subject math problems with diagrams, each transformed into six distinct versions, totaling 15K test samples.*

- **MathVision**: *A meticulously curated collection of 3,040 high-quality mathematical problems with visual contexts sourced from real math competitions. Spanning 16 distinct mathematical disciplines and graded across 5 levels of difficulty.*

- **OlypamidBench**: *A dataset comprising 8,476 math and physics problems sourced from International Olympiads, Chinese Olympiads, and the Chinese College Entrance Exam (GaoKao). It features expert-level annotations for step-by-step reasoning.*

- **WeMath**: *Inspired by human-like mathematical reasoning, WeMath is the first benchmark specifically designed to explore problem-solving principles beyond end-to-end performance. It meticulously collects and categorizes 6.5K visual math problems, spanning 67 hierarchical knowledge concepts and 5 layers of knowledge granularity.*

- **MMK12_test**: *A dataset focusing on subject-specific tasks in physics, chemistry, biology, and mathematics. It includes questions with verified answers and is designed to evaluate subject-specific reasoning capabilities.*

### A.2    BASELINES

Following the taxonomy used in Table 1, we group baseline systems into three categories:

- **Closed-source proprietary MLLMs**: Claude3.7-Sonnet, GPT-4o, o1, Gemini2-flash.
- **Open-source base MLLMs**: InternVL2.5-VL-8B, Qwen-2.5-VL-7B, InternVL2.5-VL-38B, InternVL2.5-VL-78B.
- **Open-source fine-tuned MLLMs**: InternVL2.5-8B-MPO, InternVL2.5-38B-MPO, QVQ-72B-Preview, Adora-7B, R1-Onevision-7B, OpenVLThinker-7B, MM-Eureka-7B, R1-ShareVL-7B.

Our proposed system, **DIVA-GRPO-7B**, is instantiated on the same backbone as Qwen2.5-VL-7B for fair comparison.

## B    THEORETICAL ANALYSIS OF ADVANTAGE NORMALIZATION AND DIFFICULTY-WEIGHTED BALANCING

**Theorem B.1** (Gradient Variance Control). *Let $g(\theta)$ be the stochastic policy gradient estimator*

$$g(\theta) = \sum_{t=1}^{T} A_t \, \nabla_\theta \log \pi_\theta(a_t|s_t),$$

*where $A_t$ denotes the advantage function. Suppose $\mathbb{E}[g(\theta)] = \nabla J(\theta)$ is unbiased and $Var[g(\theta)] < \infty$. Then for step size $\eta > 0$, the expected squared error satisfies*

$$\mathbb{E}\big[\|\theta_{t+1} - \theta^*\|^2\big] \leq \mathbb{E}\big[\|\theta_t - \theta^*\|^2\big] - \eta\|\nabla J(\theta_t)\|^2 + \eta^2 \, Var[\tilde{g}(\theta_t)]. \tag{1}$$

*This inequality shows that the convergence rate depends critically on the variance of the gradient estimator. Reducing gradient variance improves optimization stability and accelerates convergence.*

**Corollary B.2** (Advantage Normalization and Difficulty-Weighted Balancing). *Consider the adjusted gradient estimator*

$$\tilde{g}(\theta) = \sum_{i=1}^{k} \hat{A}(y_i) \nabla_\theta \log \pi_\theta(y_i),$$

*where the adjusted advantage is defined as*

$$\hat{A}(y_i) = \exp\Big( k \cdot (D_i - \bar{D}) \cdot \text{sgn}(\tilde{A}(y_i)) \Big) \tilde{A}(y_i).$$

*Here $\tilde{A}(y_i)$ is the batch-level normalized advantage (either local or global), $D_i$ is the difficulty coefficient of the variant $q^{(i)}$, $\bar{D}$ is the mean difficulty within the group, and $k > 0$ is a sensitivity parameter.*

*Then:*

1. **Unbiasedness:** *Normalization enforces zero-mean scaling, so $\mathbb{E}[\tilde{g}(\theta)] = \nabla J(\theta)$.*

2. **Variance Reduction:** *Difficulty-weighted scaling adaptively reweights samples. Hard problems amplify correct signals and soften incorrect ones, while easy problems down-weight correct signals and penalize incorrect ones more strongly. This rebalancing prevents domination by outliers, ensuring*

$$Var[\tilde{g}(\theta)] \leq Var[g(\theta)].$$

*Substituting into Theorem B.1 yields*

$$\mathbb{E}\big[\|\theta_{t+1} - \theta^*\|^2\big] \leq \mathbb{E}\big[\|\theta_t - \theta^*\|^2\big] - \eta\|\nabla J(\theta_t)\|^2 + \eta^2 \, Var[\tilde{g}(\theta_t)]. \tag{2}$$

*Hence, the proposed balancing strategy preserves unbiased gradient estimates while reducing update variance, leading to more stable and efficient convergence.*

**Discussion.** Theorem B.1 establishes that variance directly governs the convergence behavior of policy gradient methods. Corollary B.2 demonstrates that our difficulty-weighted and normalized advantage balancing reduces variance while maintaining unbiasedness. Intuitively, this balances contributions across problem difficulties and prevents either global or local advantages from dominating optimization, thereby stabilizing training and improving efficiency.

## C  THEORETICAL PROOF OF OPTIMAL REWARD BALANCE UNDER Z-SCORE NORMALIZATION

To further justify our design choices, we provide a theoretical proof. Our method adaptively assesses the difficulty of each problem and generates variants to keep the overall difficulty moderate for the current model, aiming to maintain an approximately 50/50 ratio of correct and incorrect samples (rather than relying on a single correct or incorrect sample to produce an advantage). For simplicity, we consider a binary-reward setting, where correct answers receive the maximum reward and incorrect answers receive zero. The proof shows that under this balanced setting, optimization is most efficient: when correct and incorrect samples are equally represented, the estimated gradient aligns most closely with the optimal update direction, leading to more effective model improvement.

**Setup and notation.** Consider a single batch (or mini-batch) of $n$ rollouts in a GRPO-like procedure. Rewards take only two values $\{0, R_{\max}\}$. Let $k$ be the number of rollouts with reward $R_{\max}$ and set

$$\mu := \frac{k}{n} \in [0, 1].$$

We compute advantages by z-score normalizing rewards over the batch:

$$A_i = \frac{r_i - \bar{r}}{\sigma_r}, \qquad \bar{r} = \mu R_{\max}, \qquad \sigma_r = \sqrt{\text{Var}(r)} = R_{\max}\sqrt{\mu(1-\mu)}.$$

Let $g_i := \nabla_\theta \log \pi(a_i|s_i) \in \mathbb{R}^d$ denote the per-sample score-gradient vector. We assume the parameter update direction (up to a positive constant) follows the common policy-gradient form

$$\Delta\theta \propto \frac{1}{n} \sum_{i=1}^{n} A_i\, g_i.$$

Fix a reference unit vector $v \in \mathbb{R}^d$ that represents the desired "optimal" direction. Define

$$s_+ := v^\top \mathbb{E}[g \mid r = R_{\max}], \qquad s_- := v^\top \mathbb{E}[g \mid r = 0],$$

the average projection of class gradients onto $v$.

**Lemma C.1** (advantages for binary rewards). *In the above notation the two advantage values are*

$$A_+ := \frac{R_{\max} - \bar{r}}{\sigma_r} = \sqrt{\frac{1-\mu}{\mu}}, \qquad A_- := \frac{0 - \bar{r}}{\sigma_r} = -\sqrt{\frac{\mu}{1-\mu}}.$$

*In particular, $R_{\max}$ cancels and does not affect the dependence on $\mu$.*

*Proof.* Direct substitution yields

$$A_+ = \frac{R_{\max} - \mu R_{\max}}{R_{\max}\sqrt{\mu(1-\mu)}} = \frac{1-\mu}{\sqrt{\mu(1-\mu)}} = \sqrt{\frac{1-\mu}{\mu}},$$

$$A_- = \frac{0 - \mu R_{\max}}{R_{\max}\sqrt{\mu(1-\mu)}} = -\frac{\mu}{\sqrt{\mu(1-\mu)}} = -\sqrt{\frac{\mu}{1-\mu}}.$$

$\square$

**Lemma C.2** (batch update projection onto $v$). *Under the class-mean approximation (aggregating samples of each reward class into their class means), the projection of the update onto $v$ satisfies (up to a positive constant)*

$$v^\top \Delta\theta \;\propto\; \mu A_+ s_+ + (1-\mu)A_- s_- = \sqrt{\mu(1-\mu)}\,(s_+ - s_-).$$

*Hence, the absolute projected magnitude is*

$$\left|v^\top \Delta\theta\right| \;\propto\; \sqrt{\mu(1-\mu)}\,|s_+ - s_-|.$$

*Proof.* Aggregate the sum by reward classes:

$$\Delta\theta \propto \mu A_+ \, \mathbb{E}[g \mid r = R_{\max}] + (1-\mu)A_- \, \mathbb{E}[g \mid r = 0].$$

Projecting onto $v$ and substituting the expressions for $A_\pm$ gives

$$v^\top \Delta\theta \propto \mu A_+ s_+ + (1-\mu)A_- s_- = \sqrt{\mu(1-\mu)}\,s_+ - \sqrt{\mu(1-\mu)}\,s_- = \sqrt{\mu(1-\mu)}\,(s_+ - s_-).$$

Taking absolute value yields the stated expression. $\square$

**Theorem C.3** (optimality of $\mu = \frac{1}{2}$). *For any fixed $s_+, s_- \in \mathbb{R}$, the absolute projected update magnitude*

$$F(\mu) := \sqrt{\mu(1-\mu)}\,|s_+ - s_-|$$

*over $\mu \in [0,1]$ attains its maximum at $\mu = \frac{1}{2}$. Therefore, under the stated assumptions, a batch with half the samples having reward $R_{\max}$ and half having reward $0$ maximizes the expected update magnitude in direction $v$.*

*Proof.* Since $|s_+ - s_-|$ is independent of $\mu$, it suffices to maximize $g(\mu) := \sqrt{\mu(1-\mu)}$ on $[0,1]$. Note

$$g(\mu)^2 = \mu(1-\mu) = -\mu^2 + \mu,$$

a concave quadratic with vertex at $\mu = \frac{1}{2}$, where it attains the maximum value $\frac{1}{4}$. Hence $g(\mu)$ is maximized at $\mu = \frac{1}{2}$, and so is $F(\mu)$. $\square$

**Two representative geometric cases.** Here we discuss two extreme yet representative cases separately.

**Corollary C.4** (Case A: opposite-class gradients). *Assume that the average class gradients are exactly opposite along the reference direction, i.e.*

$$s_- = -s_+.$$

*Then the projected update satisfies (up to a positive constant)*

$$v^\top \Delta\theta \propto \sqrt{\mu(1-\mu)}\,(s_+ - s_-) = 2\sqrt{\mu(1-\mu)}\,s_+,$$

*and hence the absolute projected magnitude is proportional to*

$$\left|v^\top \Delta\theta\right| \propto 2|s_+|\sqrt{\mu(1-\mu)}.$$

*Consequently this magnitude is maximized at $\mu = \frac{1}{2}$.*

*Proof.* Substitute $s_- = -s_+$ into the general expression $v^\top \Delta\theta \propto \sqrt{\mu(1-\mu)}(s_+ - s_-)$. This yields the displayed expression, which is a scalar multiple of $\sqrt{\mu(1-\mu)}$. The factor $\sqrt{\mu(1-\mu)}$ is maximized at $\mu = \frac{1}{2}$, hence the result. $\square$

**Corollary C.5** (Case B: orthogonal-class gradients). *Suppose the class-mean gradient for the negative class is orthogonal to the reference direction, i.e.*

$$s_- = 0,$$

*while $s_+ \neq 0$. Then*

$$v^\top \Delta\theta \propto \sqrt{\mu(1-\mu)}\,s_+,$$

*so the absolute projected magnitude is*

$$\left|v^\top \Delta\theta\right| \propto |s_+|\sqrt{\mu(1-\mu)},$$

*which is again maximized at $\mu = \frac{1}{2}$.*

*Moreover, consider the stronger geometric picture where the mean gradient of the positive class lies along $v$ with norm $\|g_+\| = \alpha$ and the mean gradient of the negative class is orthogonal with norm $\|g_-\| = \beta$. If the normalized advantage-weighted contributions from the two classes have equal amplitudes (i.e. $\alpha\sqrt{\mu(1-\mu)} = \beta\sqrt{\mu(1-\mu)}$, equivalently $\alpha = \beta$), then the resulting update vector is the sum of two orthogonal vectors of equal magnitude, so the angle $\phi$ between the update and $v$ satisfies $\cos\phi = 1/\sqrt{2}$ (independent of $\mu$), while the absolute magnitude of the projection onto $v$ still scales as $\sqrt{\mu(1-\mu)}$ and is maximized at $\mu = \frac{1}{2}$.*

*Proof.* The first statement follows immediately from substituting $s_- = 0$ into the general projection formula. For the geometric remark, if the two (orthogonal) class-mean vectors have equal weighted amplitudes, their vector sum has length $\sqrt{2}$ times one amplitude, and the projection onto $v$ equals that amplitude; dividing by the total norm gives $\cos\phi = 1/\sqrt{2}$. The dependence on $\mu$ appears only through the common amplitude factor $\sqrt{\mu(1-\mu)}$, which is maximized at $\mu = \frac{1}{2}$. $\square$

**Remarks.**

1. The conclusion does not depend on $R_{\max}$: z-score standardization cancels the reward scale.

2. If $s_+ > s_-$ then $v^\top \Delta\theta > 0$ and the update moves toward $v$; if $s_+ < s_-$ the update moves opposite to $v$. In either case the absolute magnitude of the projection is maximal at $\mu = \frac{1}{2}$.

3. The proof uses a simplifying class-mean approximation. In practice each class has internal variance; nonetheless the leading-order dependence of the expected projection on $\mu$ is governed by $\sqrt{\mu(1-\mu)}$. Under mild conditions on within-class variance, the qualitative conclusion (maximum at $\mu = \frac{1}{2}$) remains valid in expectation.

4. For non-binary rewards the algebra changes: one must analyse $\frac{r_i - \bar{r}}{\sigma_r}$ for the full reward distribution. The binary case provides a clear analytic baseline and shows the key role of z-score normalization in making the effective signal scale proportional to $\sqrt{\mu(1-\mu)}$.

| Variant | Description |
|---------|-------------|
| 1 | Gaussian (intensity = 0.45) and Rotate (multiples of 1°) + Variant text / Add text to image |
| 2 | Gaussian (intensity = 0.45) or Rotate (multiples of 30°) + Variant text / Add text to image |
| 3 | Gaussian (intensity = 0.3) or Rotate (multiples of 45°) + Variant text / Add text to image |
| 4 | Gaussian (intensity = 0.3) or Rotate (multiples of 90°) |
| 5 | Variant text |
| 6 | Original query / Variant text |
| 7 | 1-think step |
| 8 | 2-think steps |
| 9 | 3-think steps |

Table C.1: Variants from Hard to Easy.

Table C.2: Performance comparison across different base model.

| Model | Mathematics | Physics | Chemistry | Biology | Avg. |
|-------|-------------|---------|-----------|---------|------|
| **Qwen3-VL-4B** | | | | | |
| Qwen3-VL-4B-Instruct | 65.5 | 55.5 | 58.5 | 56.1 | 58.9 |
| +GRPO | 75.5 | 68.5 | 71.0 | 68.6 | 70.9 |
| **+DIVA-GRPO (Ours)** | 78.3 | 70.7 | 72.7 | 71.9 | 73.4 |
| improv. | +2.8 | +2.2 | +1.7 | +3.3 | +2.5 |
| **Qwen3-VL-8B** | | | | | |
| Qwen3-VL-8B-Instruct | 71.5 | 64.5 | 66.5 | 65.5 | 67.0 |
| +GRPO | 77.5 | 72.5 | 74.0 | 73.2 | 74.3 |
| **+DIVA-GRPO (Ours)** | 82.2 | 74.3 | 78.2 | 79.3 | 78.5 |
| improv. | +4.7 | +1.8 | +4.2 | +6.1 | +4.2 |

**Conclusion.** This theoretical analysis rigorously justifies the core motivation of our method. When batch advantages are computed via z-score normalization and rewards are binary $\{0, R_{\max}\}$, the expected magnitude of the update projected onto any fixed reference direction $v$ is proportional to $\sqrt{\mu(1 - \mu)}$, which attains its maximum when $\mu = \frac{1}{2}$. In other words, a batch balanced with equal numbers of correct and incorrect samples provides the strongest expected directional signal for optimization. This result supports our design choice of adaptively generating problem variants to maintain an approximately 50/50 ratio of correct and incorrect samples, ensuring that gradient updates most efficiently align with the optimal update direction and thereby maximizing model improvement.

# D  SUPPLEMENTARY EXPERIMENTS

## D.1  PARAMETER $k$

In the same experimental setup as the ablation study, we assess the performance of the proposed method across different values of the parameter $k$. As shown in Figure D.1a, the model achieves optimal performance when $k \approx 0.1$, with a peak accuracy difference of only 0.5%. When $k = 0$ (i.e., difficulty weighting is disabled) or $k = 1$ (i.e., excessive difficulty weighting), the algorithm exhibits similar performance in the early training stages due to the lack of distinction in sample difficulties. However, as training progresses, the differences in sample difficulties become more pronounced, and overly extreme weighting results in degraded performance or training instability. When $k = 0.1$, the maximum weight applied is roughly 1.5 times the original advantage, while the maximum reduction is about 0.67 times the original value. From the weighting formula, we derive $k = \frac{\ln(1.5)}{\max\left(D_q^{(i)} - \bar{D}_q\right)}$.

## D.2  PARAMETER $\eta$ AND SAMPLE SAMPLING

In the same experimental setup as the ablation study, we evaluate the performance of the proposed method across different values of the parameter $\eta$. The parameter $\eta$ represents the magnitude of change in the difficulty assessment when a question is answered correctly or incorrectly. Specifically, when $\eta = 2$, the sample requires four consecutive correct (or incorrect) answers before its difficulty level is updated to the easiest (or hardest) level. When $\eta = 4$, two correct (or incorrect) answers are required for the update, and when $\eta = 8$, only one correct (or incorrect) answer is needed for the update.

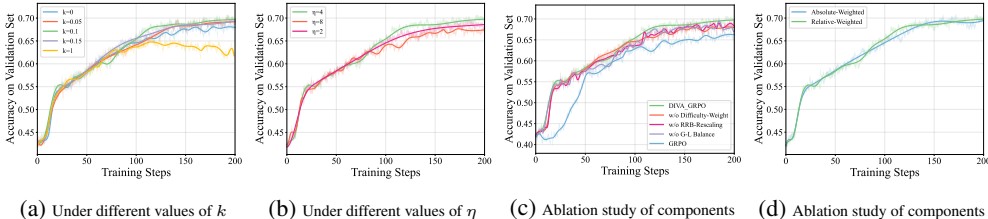

(a) Under different values of $k$    (b) Under different values of $\eta$    (c) Ablation study of components    (d) Ablation study of components

Figure D.1: Effect of different hyperparameters on model performance on the validation set.

Table D.3: Performance comparison across different external models on Qwen-2.5-VL-7B.

| Model | Mathematics | Physics | Chemistry | Biology | Avg. |
|---|---|---|---|---|---|
| **Qwen-2.5-VL-7B** | | | | | |
| + Qwen-Plus | 78.1 | 61.5 | 69.1 | 70.9 | 69.8 |
| + GPT-o3 | 78.3 | 62.2 | 69.6 | 70.7 | 70.2 |
| **improv.** | +0.2 | +0.7 | +0.5 | -0.2 | +0.3 |

We observe that when the difficulty assessment changes too rapidly, the inherent randomness in the model may result in inaccurate difficulty assessments. Conversely, when the difficulty assessment changes too slowly, the model's performance improvement may not be reflected in the difficulty evaluations. As shown in Figure D.1b, the model achieves the best performance when $\eta = 4$. Based on the difficulty update formula, we suggest that $\eta$ should be set as $\frac{D_{\max} - D_{\min}}{2}$.

Additionally, regarding sample sampling, we configure a total of nine difficulty level variants, as detailed in Table C.1. For sampling, we use a normal distribution to sample from the difficulty level adjacent to the target difficulty.

## D.3 MODELS OF DIFFERENT SCALES

As shown in Table C.2, we evaluate the performance of GRPO and our method, DIVA-GRPO, on different model sizes and architectures, with the experimental setup consistent with the one described earlier. The experimental results on Qwen3-VL-4B-Instruct and Qwen3-VL-8B-Instruct show that, compared to the Base model, our method improves by 14.5 and 11.5, respectively. Additionally, our method achieves an average improvement of 2.5 and 4.2 over GRPO. These results demonstrate that our approach is effective across various model sizes and architectures.

## D.4 GENERATION USING DIFFERENT EXTERNAL MODELS

As shown in Table D.3, we generate variants and think steps using Qwen-plus and GPT-o3, and test the performance they achieve under the same training conditions. The results show that the impact of using a strong or weaker external model on the final model training performance is minimal, with GPT-o3 achieving only a 0.3 accuracy improvement over Qwen-plus. This can be attributed to our use of result supervision, where only samples that can correctly answer the final result are selected for training, rather than training on all samples. Additionally, we validate the consistency of the generated question variants with the original questions using Qwen-plus. We sample 1,000 question variants for the experiment and find that 98.8% of the variants are consistent with the original questions. This further demonstrates the reliability of our method. The prompts used in the experiments are shown in Section G.

## D.5 ABSOLUTE DIFFICULTY WEIGHTING VS. RELATIVE DIFFICULTY WEIGHTING

As described in Section 3.3, our method uses relative difficulty weighting instead of absolute difficulty weighting. This is because the difficulty of a problem and its variants differ across two dimensions. For example, a problem with difficulty 1 plus a variant with difficulty 8 is not directly comparable to a problem with difficulty 9. The variant with increased difficulty simply adds some visual noise, while the problem with inherent difficulty is challenging due to its complex logic. Therefore, applying absolute difficulty weighting intuitively does not make sense. However, for variants of the same original problem, the difficulty ranking is clear: image-noise variant > original problem > variant

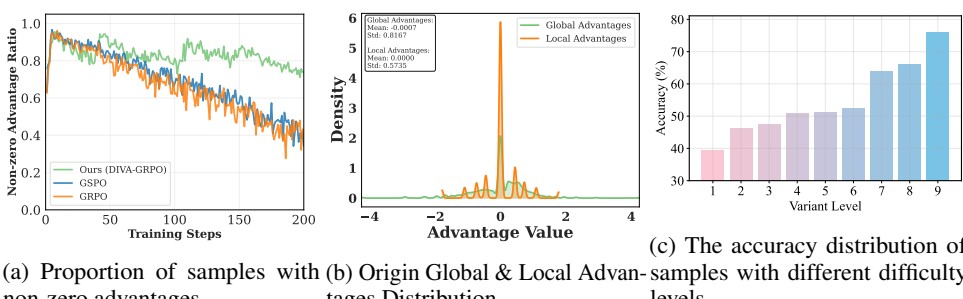

(a) Proportion of samples with non-zero advantages

(b) Origin Global & Local Advantages Distribution

(c) The accuracy distribution of samples with different difficulty levels.

Figure D.2: Analysis of the Effectiveness of DIVA-GRPO.

with added think-steps. As a result, we adopt relative difficulty weighting, where the difficulty within the same problem can be compared. Additionally, as shown in Figure D.1d, we test both absolute and relative difficulty weighting and find that their impact on the model's final performance is minimal. However, from the perspective of logical consistency, we ultimately choose relative difficulty weighting.

## D.6 ANALYSIS OF THE EFFECTIVENESS OF THE METHOD

**Proportion of Samples with Non-zero Advantage.** Our method dynamically estimates problem difficulty and samples variants of appropriate difficulty, effectively mitigating reward sparsity and improving sample efficiency. As shown in Figure D.2a, the proportion of non-zero advantage samples in GRPO and GSPO rises rapidly at the beginning of training, indicating a quick increase in training accuracy. However, as the model becomes stronger, most problems become easy, causing a sharp drop in the proportion of non-zero advantage samples and limiting late-stage optimization. In contrast, our method adaptively adjusts problem difficulty, keeping the proportion of non-zero advantage samples stable, ensuring the model continually encounters challenging variants, maintains a balanced reward distribution, and sustains effective training throughout.

**Analysis of Origin Local & Global Advantages.** We first apply z-score normalization within each group for both Local and Global Advantage. Due to the higher sparsity and variance in the Global group compared to the Local group, the range of advantages,$\max(\mathrm{adv}) - \min(\mathrm{adv})$ can still differ significantly between groups even after standard normalization, causing the group with the larger range to dominate (see Figure D.2b). To mitigate this effect, we further apply batch normalization separately to the Local and Global advantages, followed by standard normalization within each batch. This additional normalization effectively reduces the discrepancy between the two distributions.

**Analysis of Variant Difficulty.** To validate that the difficulty of the variants generated by our method aligns with expectations, we randomly sample 500 examples from the MMK12_test dataset. For each example, we generate one variant at each difficulty level, with the specific variant settings detailed in Table C.1. This results in 500 samples per difficulty level. For each sample, we perform 10 rollouts using Qwen-2.5-VL-7B and compute the final accuracy. The distribution of the final accuracy is shown in Figure D.2c. Although the increase in accuracy is not perfectly linear, the overall trend indicates that our sample variants exhibit the expected pattern: easier variants have higher accuracy, and more difficult variants have lower accuracy. This demonstrates the effectiveness of our method.

Table F.4: Performance comparison across different disciplines in MMK12. Bold denotes the best performance among 7B models, and underline marks the best overall performance.

| Model | Mathematics | Physics | Chemistry | Biology | Avg. |
|---|---|---|---|---|---|
| **Closed-Source Models** | | | | | |
| Claude3.7-Sonnet | 57.4 | 53.4 | 55.4 | 55.0 | 55.3 |
| GPT-4o | 55.8 | 41.2 | 47.0 | 55.4 | 49.9 |
| o1 | 81.6 | 68.8 | 71.4 | 74.0 | 73.9 |
| Gemini2-flash | 76.8 | 53.6 | 64.6 | 66.0 | 65.2 |
| **Open-Source General Models** | | | | | |
| InternVL2.5-VL-8B | 46.8 | 35.0 | 50.0 | 50.8 | 45.6 |
| Qwen-2.5-VL-7B | 58.4 | 45.4 | 56.4 | 54.0 | 53.6 |
| InternVL2.5-VL-38B | 61.6 | 49.8 | 60.4 | 60.0 | 58.0 |
| Qwen-2.5-VL-32B | 71.6 | 59.4 | 69.6 | 66.6 | 66.8 |
| InternVL2.5-VL-78B | 59.8 | 53.2 | 68.0 | 65.2 | 61.6 |
| Qwen-2.5-VL-72B | 75.6 | 64.8 | 69.6 | 72.0 | 70.5 |
| **Open-Source Reasoning Models** | | | | | |
| InternVL2.5-8B-MPO | 26.6 | 25.0 | 42.4 | 44.0 | 34.5 |
| InternVL2.5-38B-MPO | 41.4 | 42.8 | 55.8 | 53.2 | 48.3 |
| QVQ-72B-Preview | 61.4 | 57.4 | 62.6 | 64.4 | 61.5 |
| Adora | 63.6 | 50.6 | 59.0 | 59.0 | 58.1 |
| R1-Onevision | 44.8 | 33.8 | 39.8 | 40.8 | 39.8 |
| OpenVLThinker | 63.0 | 53.8 | 60.6 | 65.0 | 60.6 |
| MM-Eureka-7B | 71.2 | 56.2 | 65.2 | 65.2 | 64.5 |
| **DIVA-GRPO-7B (Ours)** | **78.3** | **62.2** | **69.6** | **70.7** | **70.2** |

## E    EFFECT OF QUESTION DIFFICULTY ON GRPO OPTIMIZATION

To further validate that providing the model with moderately difficult questions is more conducive to optimization via GRPO, we conducted a new experiment. Two sets of 5,000 samples were drawn from the MMK12 training data: the first set consisted of randomly sampled instances, while the second set comprised instances of moderate difficulty. Both sets were used to train the model under standard GRPO with a rollout parameter of $k = 5$. As shown in Figure E.3, training on the moderately difficult data enabled faster and more effective model optimization.

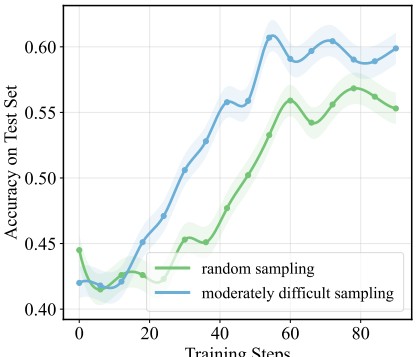

Figure E.3: Comparison of GRPO optimization on randomly sampled versus moderately difficult questions.

## F    CROSS-DISCIPLINARY PERFORMANCE ON MMK12

For the MMK12 dataset, we observed an interesting phenomenon: although our model is primarily trained on mathematics problems, it also demonstrates improved reasoning capabilities in physics, chemistry, and biology. The experimental results are summarized in Table F.4. Our model achieves the best performance among 7B models across all four disciplines and even surpasses many closed-source commercial and open-source general large models. Comparative results for other models can be found in Meng et al. (2025).

## G    PROMPTS USED IN EXPERIMENTS

In this section, we provide the detailed prompts used in our pipeline, including variant generation, reasoning, and verification.

## G.1 PART I: VARIANT GENERATION

**Purpose:** Generate 5 distinct variants of a problem while preserving the exact same correct answer. Variants must include the token `<image>` and be wrapped in `<variant*>...</variant*>` tags.

PROMPT 1: VARIANT GENERATION TEMPLATE

```
Generate 5 distinct variants of the following problem that:
1. Preserve the exact same correct answer as the original.
2. Use significantly different wording, sentence structure.
3. You can adjust the sentence length--either making it concise (using
    streamlined language) or extending it (by explaining the question
    content in detail or complicating it with advanced language to
    increase difficulty)--but must ensure correctness.
4. Format of variants as, include <image> in the variants:
<variant1>[First variant's full text]</variant1>
<variant2>[Second variant's full text]</variant2>
...
<variant5>[Fifth variant's full text]</variant5>
Original Problem:
{Original_Problem_Text}
```

**Expected output (example):**

```
<variant1>[... text ... <image> ...]</variant1>
<variant2>[... text ... <image> ...]</variant2>
...
<variant5>[... text ... <image> ...]</variant5>
```

## G.2 PART II: REASONING AND VERIFICATION

PROMPT 2: STEP-BY-STEP REASONING (`PROMPT_FORMAT_THINK_STEP`)

**Purpose:** Generate a stepwise reasoning chain (3–5 steps) with detailed diagram observations, ending with the final answer boxed.

```
You are a mathematician, statistician, and geometer. Below, I will
    present you with a math problem along with its accompanying diagram.
    Please carefully observe the details in the image.
Given the text, images, generate a step-by-step reasoning process that
    logically leads to the correct result in the \boxed{}. Requirements:
    Flexible step count (3-5 steps)...
[full problem text and image here]
```

**Expected output (example):**

```
<step1> ... observation ... </step1>
<step2> ... inference ... </step2>
<step3> ... inference ... </step3>
<step4> ... conclusion ... </step4>
\boxed{[final answer]}
```

PROMPT 3: ANSWER COMPARISON

**Purpose:** Compare two numerical answers (`answer1`, `answer2`) ignoring formatting. Returns `<answer>True</answer>` if equal, else `False`.

```
Compare these two answers numerically, ignoring any formatting
    differences:
Answer 1: {result['answer1']}
Answer 2: {result['answer2']}

Extract just the numerical values from each answer and compare them. If
    the numerical values are the same, return True. Otherwise return
    False. Reply with <answer>True</answer> or <answer>False</answer>
```

PROMPT 4: REFLECTION AND REVISION (PROMPT_FORMAT_THINK_STEP_2)

**Purpose:** Refine reasoning steps after receiving correctness feedback, without directly inserting the correct answer in the steps or reverse-engineering from it.

```
You are a mathematician, statistician, and geometer. Please carefully
    observe the details in the image.
You have already provided the reasoning steps and correct answer above.
Do you think your answer is correct? Revise or improve your reasoning
    steps based on the correct answer.
Emphasize!
- DO NOT include the correct answer in <step>.
- DO NOT reverse-engineer from the answer.
[full problem text and image here]
```

# H  CASE STUDY OF TRAIN DATASET

## H.1  CASE 1

**Problem:**  The figure <image> below shows the graphs of the functions: $y = x^2 - 1$, $y = x^2 + 6x + 8$, $y = x^2 - 6x + 8$, $y = x^2 - 12x + 35$ in the same Cartesian coordinate system. The most likely graph for $y = x^2 - 6x + 8$ is ___.

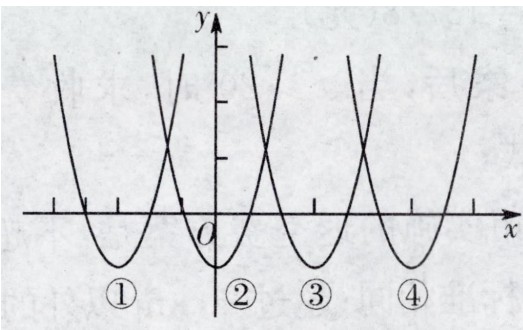

Figure H.4: Original Problem

DIFFICULT VARIANT

**Text and Image Variant:**  Embed both the original problem description and the question directly into the image, and replace the prompt text with the following fixed instruction:

*As shown in the <image>, answer the question according to the figure.*

**Image Variants:**

- **Salt Noise:** The image is perturbed with salt noise to increase complexity in recognizing the object.
- **Blur:** The image is blurred, making it harder to focus on specific details.
- **Gauss Noise:** The image is perturbed with Gaussian noise.

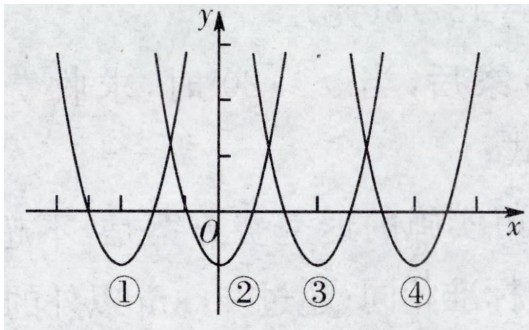

The figure <image> below shows the graphs of the functions: $$y=x^{2}-1$$, $$y=x^{2}+6x+8$$, $$y=x^{2}-6x+8$$, $$y=x^{2}-12x+35$$ in the same Cartesian coordinate system. The most likely graph for $$y=x^{2}-6x+8$$ is ___.

Figure H.5: Vision-Dominant Problem

- **Rotate:** The image is rotated, increasing difficulty in spatial reasoning.

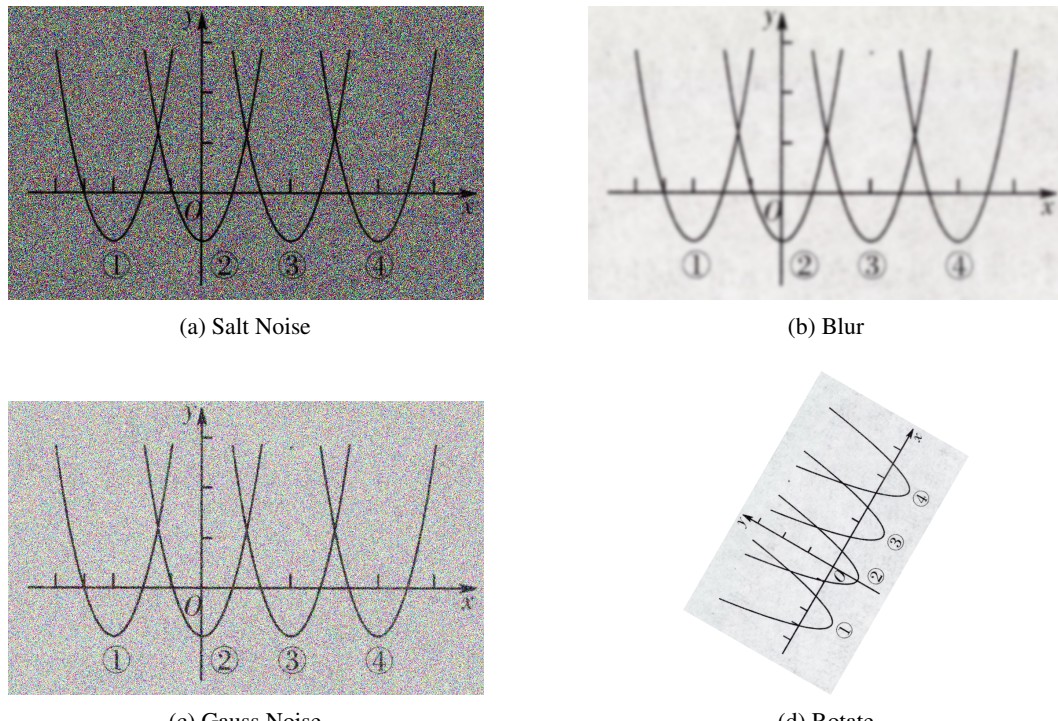

(a) Salt Noise

(b) Blur

(c) Gauss Noise

(d) Rotate

Figure H.6: Image Variants: Salt Noise, Blur, Gauss Noise, and Rotate.

**Text Variants:** Semantically equivalent reformulations of the original problem, preserving both meaning and the correct answer.

**Problem Variant 1:**

In the Cartesian plane depicted in the figure <image>, four distinct parabolic curves are illustrated, each corresponding to one of the following equations: $y = x^2 - 1$, $y = x^2 + 6x + 8$, $y = x^2 - 6x + 8$, and $y = x^2 - 12x + 35$. Your challenge is to discern and point out the specific curve that is generated by the equation $y = x^2 - 6x + 8$.

**Problem Variant 2:**

Among the provided graphs in the figure `<image>`, identify which one corresponds to the quadratic function $y = x^2 - 6x + 8$. This function, when compared to the others—$y = x^2 - 1$, $y = x^2 + 6x + 8$, and $y = x^2 - 12x + 35$—displays a unique vertex and orientation. Your task is to select the graph that accurately represents $y = x^2 - 6x + 8$.

EASIER VARIANT

**Think Step:**

- **Step 1:** The image displays four parabolic graphs labeled ①, ②, ③, and ④. Each graph represents a quadratic function of the form $y = ax^2 + bx + c$. The general characteristics of these parabolas include their orientation (all open upwards, indicating a positive leading coefficient) and their respective vertices.

- **Step 2:** To identify which graph corresponds to the function $y = x^2 - 6x + 8$, we first rewrite this function in vertex form. Completing the square:

$$y = x^2 - 6x + 8 = (x^2 - 6x + 9) - 9 + 8 = (x - 3)^2 - 1$$

This reveals that the vertex of the parabola is at the point $(3, -1)$.

- **Step 3:** Next, we analyze the positions of the vertices for each graph:
    - Graph ① has its vertex to the left of the y-axis.
    - Graph ② has its vertex at the origin or very close to it.
    - Graph ③ has its vertex at $(3, -1)$, as determined from the vertex form.
    - Graph ④ has its vertex to the right of the y-axis but not at $(3, -1)$.

- **Step 4:** Thus, the graph with the vertex at $(3, -1)$ is graph ③. Given the analysis, the function $y = x^2 - 6x + 8$ corresponds to the graph labeled ③, as it is the only graph with the correct vertex coordinates.

## H.2 CASE 2

**Problem:**

`<image>` Subtract all brown cubes. Subtract all blue cylinders. How many cubes are left?

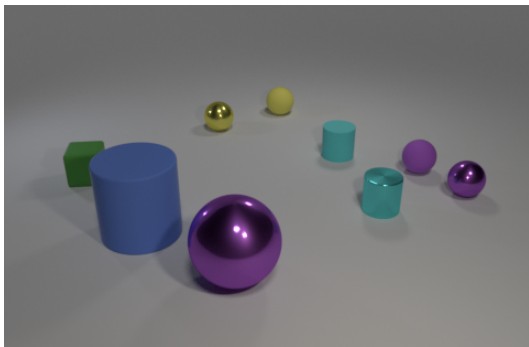

Figure H.7: Original Problem

DIFFICULT VARIANT

**Text and Image Variant:**  Embed both the original problem description and the question directly into the image, and replace the prompt text with the following fixed instruction:

*As shown in the `<image>`, answer the question according to the figure.*

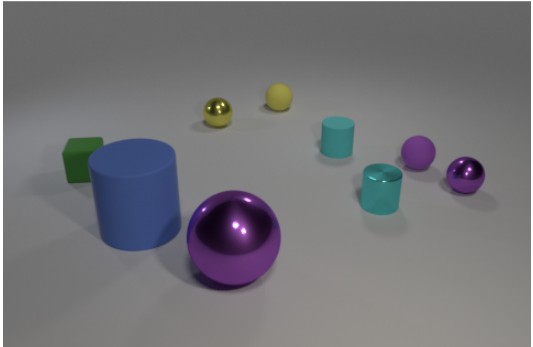

Figure H.8: Vision-Dominant Problem

**Image Variants:**

- **Salt Noise:** The image is perturbed with salt noise to increase complexity in recognizing the object.

- **Blur:** The image is blurred, making it harder to focus on specific details.

- **Gauss Noise:** The image is perturbed with Gaussian noise.

- **Rotate:** The image is rotated, increasing difficulty in spatial reasoning.

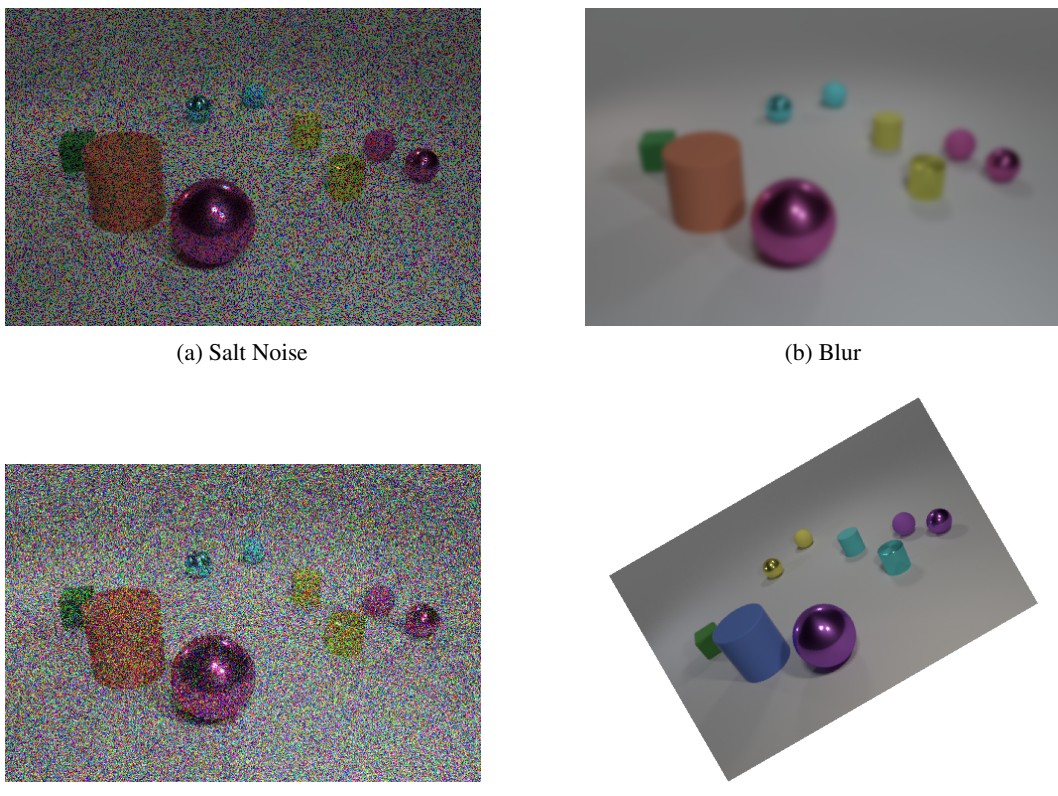

Figure H.9: Image Variants: Salt Noise, Blur, Gauss Noise, and Rotate.

**Text Variants:** Semantically equivalent reformulations of the original problem, preserving both meaning and the correct answer.

**Problem Variant 1:**

<image> Once all brown cubes and blue cylinders are taken away, what is the number of cubes left?

**Problem Variant 2:**

<image> Remove all brown cubes and all blue cylinders. How many cubes remain?

EASIER VARIANT

**Think Step:**

- **Step 1:** Leftmost object: a single cube, distinctly green.
    - Immediately to the right of that cube: a large, dark-blue cylinder.
    - Farther right: two smaller cylinders of a lighter, cyan-blue hue.
    - Several spheres (purple, gold, yellow) appear, but these are not cubes or cylinders.
    - No object in the scene exhibits a brown color.

    Thus, only one cube (green) is present, and there are three blue-family cylinders (one dark, two cyan).

- **Step 2:** Interpret the problem instructions with the observed inventory.
    - "Subtract all brown cubes." Because none exist, this operation leaves the cube count unaffected.
    - "Subtract all blue cylinders."

- **Step 3:** Although three cylinders satisfy the color condition, removing cylinders changes only the cylinder inventory; cubes remain untouched. Perform the mandated subtractions.
    - After eliminating zero brown cubes, the cube inventory is the same as noted in Step 1.

- **Step 4:** Removing the three blue cylinders affects only the cylinder category; the cube category remains exactly as before. Summarize the effect of both operations.
    - None of the specified subtractions alters the original cube count, so the final cube tally equals the initial cube tally recorded in Step 1.

# I CASE STUDY OF TEST DATASET

## I.1 CASE 1

**Problem.** As shown in the figure, $\triangle ABC \cong \triangle ADE$. From the figure we have $\angle B = 70°$, $\angle C = 30°$, and $\angle DAC = 35°$. Find $\angle EAC$.

**Choices:** A: $40°$ B: $45°$ C: $35°$ D: $25°$

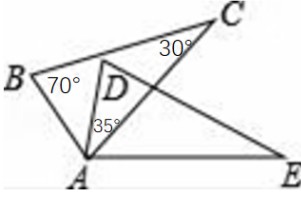

Figure I.10: Geometry problem: MathVerse 7.

OUR MODEL — FINAL ANSWER: $\boxed{\text{B: } 45°}$

**Complete derivation (external, step-by-step).**

1. In $\triangle ABC$, the triangle angle-sum gives $\angle BAC = 180° - \angle B - \angle C = 180° - 70° - 30° = 80°$.
2. By congruence $\triangle ABC \cong \triangle ADE$, corresponding angles are equal; hence $\angle DAE = \angle BAC = 80°$.
3. At vertex $A$, the angle decomposition yields $\angle DAE = \angle DAC + \angle CAE$. Given $\angle DAC = 35°$, we have $\angle CAE = 80° - 35° = 45°$.
4. Therefore $\angle EAC = \angle CAE = 45°$, corresponding to choice B.

OTHER MODEL (BASELINE) — REPORTED ANSWER: $\boxed{\text{A: } 40°}$

**Reported derivation (reconstructed from the baseline output).**

1. The baseline asserts that by congruence $\angle DAE = \angle BAC$ (this assertion is geometrically correct in form).
2. The baseline then (incorrectly) reads or treats $\angle BAC$ as $35°$ and reads $\angle CAD$ (or $\angle DAC$) as $30°$ (this is a misreading of the labels in the figure / given data).
3. Using the (incorrect) equality $\angle BAC = \angle CAD + \angle EAC$, the baseline computes $\angle EAC = 35° - 30° = 5°$.
4. The baseline then presents an alternative arithmetic attempt: it treats some combination as an "exterior" or summed angle, computing $35° + 30° = 65°$, and then (without a valid geometric justification) performs $65° - 70°$ and reports $40°$ as the final choice.

**Explicit error analysis.**

- **Label/misreading error.** The baseline's step that sets $\angle BAC = 35°$ is inconsistent with the given values $\angle B = 70°$ and $\angle C = 30°$, which imply $\angle BAC = 80°$. Misidentifying which labeled angle corresponds to $\angle BAC$ is the root cause.
- **Invalid arithmetic and unjustified operations.** The baseline mixes different angle-sum operations (interior vs. exterior) without geometric justification and carries out arithmetic (e.g., $65° - 70° \rightarrow 40°$ in the reported text) that is both algebraically incorrect and lacks geometric meaning in context.
- **Broken logical chain.** Because the baseline's premises are incorrect, subsequent deductions (such as subtracting or summing those incorrect angles) do not follow from valid geometric facts (triangle angle-sum, congruence correspondence, or correct angle decomposition).

**Summary — Why Our Model Is Preferable in This Case.**

- **Accurate visual interpretation:** Our model correctly identifies the labels and reads $\angle B$, $\angle C$, and $\angle DAC$ from the figure, avoiding the label-confusion that the baseline exhibits.
- **Correct and verifiable reasoning:** Each step follows a basic geometric rule (angle-sum in a triangle, congruence $\Rightarrow$ corresponding angles equal, and decomposition of an angle at a vertex), and the arithmetic is straightforward and checkable.
- **Transparent presentation:** The derivation above is externalized and suitable for inclusion in an appendix so that reviewers can trace the reasoning and confirm correctness.

## I.2 CASE 2

**Problem.** Write the set of numbers represented on the number line in interval notation.

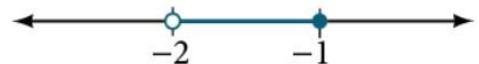

Figure I.11: Interval notation problem: MathVerse 474.

OUR MODEL — FINAL ANSWER: $\boxed{(-2, -1]}$

**Complete derivation (external, step-by-step).**

1. Inspect the endpoints shown on the number line: one endpoint is at $-2$ and the other is at $-1$.
2. Determine inclusion/exclusion from the marker style:
   - $-2$ is marked with an *open* (hollow) circle $\Rightarrow$ $-2$ is *not* included.
   - $-1$ is marked with a *solid* (filled) circle $\Rightarrow$ $-1$ *is* included.

3. Translate inclusion/exclusion to interval notation:
    - Excluding $-2$ uses a parenthesis '(' at the left.
    - Including $-1$ uses a bracket ']' at the right.
4. Therefore the interval is $(-2, -1]$.
5. In inequality form this corresponds to $-2 < x \leq -1$.

OTHER MODEL (BASELINE) — REPORTED ANSWER: $\boxed{(-2, -1)}$

**Reported derivation (reconstructed from the baseline output).**

1. The baseline identifies the endpoints as $-2$ and $-1$.
2. It (incorrectly) interprets both endpoint markers as *open* circles, concluding that neither endpoint is included.
3. From that (mis)interpretation it writes the interval using parentheses on both sides and reports $(-2, -1)$.

**Explicit error analysis.**

- **Visual misinterpretation error.** The baseline's central mistake is reading the marker at $-1$ as open when it is actually solid (closed). This single perceptual/mapping error changes '$\leq$' to '$<$' at the right endpoint and thus changes the bracket type.
- **Notation consequence.** Interval notation is sensitive to endpoint inclusion: confusing a closed endpoint for an open one converts a bracket ']' to a parenthesis ')', producing a different set. Here that error changes the set by including or excluding the point $-1$.
- **Downstream impact.** Because the baseline misreads the marker, any inequality-form statement or set-membership statement it produces (e.g., $-2 < x < -1$) will be incorrect for points equal to $-1$.

**Summary — Why Our Model Is Preferable in This Case.**

- **Correct visual-to-symbol mapping:** Our model correctly maps hollow vs. filled endpoint markers to exclusion vs. inclusion and therefore uses the appropriate parenthesis/bracket combination.
- **Clear, verifiable steps:** Each step is an elementary, checkable rule: identify endpoints, read marker style, convert to interval notation, and (optionally) supply the equivalent inequality. This transparency makes verification straightforward for reviewers.
- **Precise final expression:** The answer $(-2, -1]$ cleanly and unambiguously communicates the set of real numbers greater than $-2$ and less than or equal to $-1$, matching the figure where $-1$ is solid.

## I.3  CASE 3

**Problem.** Given the figure below, find the slope of the line.

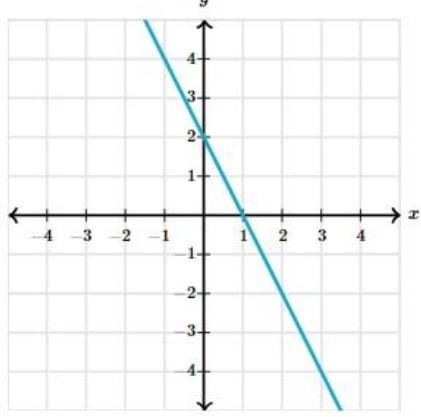

Figure I.12: Line slope problem: MathVerse 591.

OUR MODEL — FINAL ANSWER: $\boxed{-1}$

**Complete derivation (external, step-by-step).**

1. The slope of a line passing through two points $(x_1, y_1)$ and $(x_2, y_2)$ is given by:
$$\text{slope} = \frac{y_2 - y_1}{x_2 - x_1}.$$
2. From the graph, identify two points on the line: $(-1, 2)$ and $(1, 0)$.
3. Substitute these points into the slope formula:
$$\text{slope} = \frac{0 - 2}{1 - (-1)} = \frac{-2}{2} = -1.$$
4. Therefore, the slope of the line is $-1$.

OTHER MODEL (BASELINE) — REPORTED ANSWER: $\boxed{-2}$

**Reported derivation (reconstructed from the baseline output).**

1. The baseline identifies two points on the line as $(1, 2)$ and $(0, 4)$.
2. Using the slope formula:
$$\text{slope} = \frac{2 - 4}{1 - 0} = \frac{-2}{1} = -2.$$
3. The baseline reports the final slope as $-2$.

**Explicit error analysis.**

- **Incorrect point selection:** The baseline misreads the coordinates of points on the line. Using $(1, 2)$ and $(0, 4)$ does not correspond to the actual line in the figure.
- **Consequent incorrect slope:** Because the points are wrong, the computed slope $-2$ does not match the true slope of the line, which is $-1$.
- **Logical chain broken:** All subsequent reasoning is mathematically consistent with the chosen points, but the premise (point selection) is flawed, leading to an incorrect final answer.

**Summary — Why Our Model Is Preferable in This Case.**

- **Accurate visual interpretation:** Our model correctly identifies the actual points on the line from the figure.
- **Correct reasoning:** Step-by-step application of the slope formula leads directly to the correct answer $-1$.
- **Transparent derivation:** The reasoning is fully externalized, making it easy for reviewers to verify correctness and consistency with the figure.

## J CASES THAT FAIL TO GENERATE AN ADVANTAGE

### J.1 CASES THAT ARE CONSISTENTLY ANSWERED CORRECTLY

#### J.1.1 CASE 1

**Problem.** As shown in the figure, point $P$ is the intersection of the angle bisectors of the interior angle $\angle ABC$ and the exterior angle $\angle ACD$ of $\triangle ABC$. The distance from point $P$ to the line $AC$ is $6$ cm. What is the distance from point $P$ to the line $AB$?

**Answer:** $\boxed{6}$

#### J.1.2 CASE 2

**Problem.** As shown in the figure, in rhombus $ABCD$, the diagonals $AC$ and $BD$ intersect at point $O$, and $OE \parallel DC$ intersects $BC$ at point $E$. If $AD = 8$ cm, then the length of $OE$ is ____ cm.

**Answer:** $\boxed{4}$

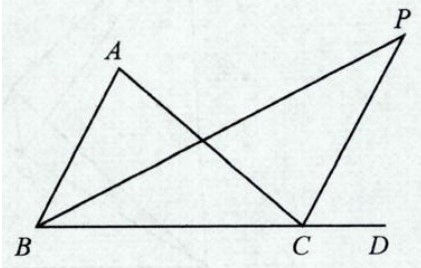

Figure J.13: Geometry problem involving angle bisectors.

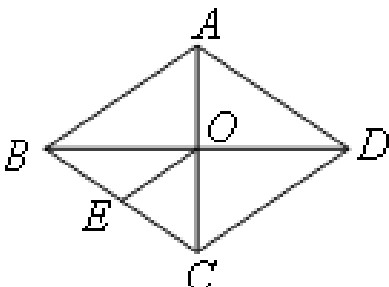

Figure J.14: Geometry problem involving a rhombus.

### J.1.3 CASE 3

**Problem.** As shown in the figure, in $\triangle ABC$, points $D$ and $E$ are on $AB$ and $AC$, respectively, and $DE \parallel BC$. If $AD = 1$ and $AB = 4$, then $\frac{DE}{BC} =$ ____.

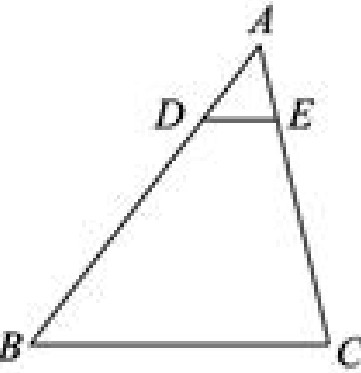

Figure J.15: Geometry problem involving similar triangles.

**Answer:** $\dfrac{1}{4}$

### J.1.4 CASE 4

**Problem.** For any non-zero real numbers $a$, $b$, if the operation principle of $a \otimes b$ is represented by the flowchart shown in the figure, then $3 \otimes 2 =$ ____.

**Answer:** $2$

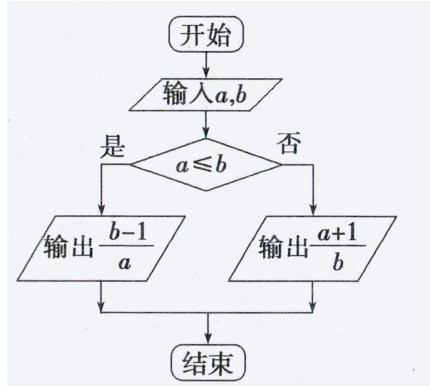

Figure J.16: Flowchart operation problem.

## J.2 CASES THAT ARE CONSISTENTLY ANSWERED INCORRECTLY

### J.2.1 CASE 1

**Problem.** Figure A is a histogram showing the frequency distribution of the monthly income of the staff and faculty of a school, where it is known that the frequency of the first group from left to right is 80. In the sample, the frequencies of the monthly income (unit: yuan) in the intervals $[1000, 1500)$, $[1500, 2000)$, $[2000, 2500)$, $[2500, 3000)$, $[3000, 3500)$, $[3500, 4000]$ are denoted as $A_1, A_2, \ldots, A_6$ respectively. Figure B is the flowchart of the algorithm used to estimate the average monthly income of the school's staff and faculty. What is the output of $S$? (Answer with a number.)

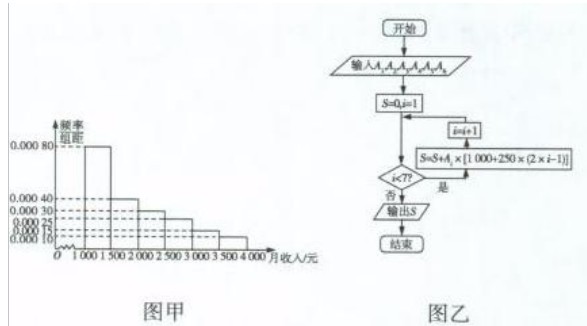

Figure J.17: Complex statistics and flowchart problem.

**Answer:** $\boxed{1962.5}$

### J.2.2 CASE 2

**Problem.** To process a part with the shape shown in the figure, calculate the degree of the inclined angle $\alpha$ based on the dimensions indicated (unit: mm). (Use a calculator, accurate to 1 second).

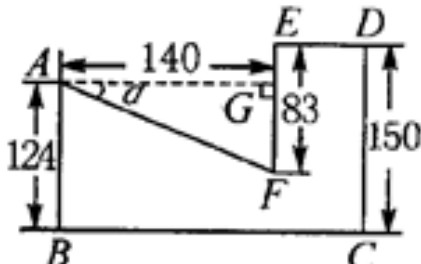

Figure J.18: Industrial geometry calculation.

**Answer:** $\boxed{22° 9' 12''}$

### J.2.3 CASE 3

**Problem.** Black and white hexagonal floor tiles are arranged according to the pattern shown in the figure; then the number of white floor tiles in the $n$-th pattern is ____.

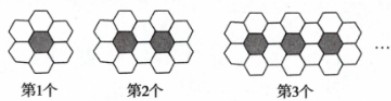

Figure J.19: Pattern recognition sequence problem.

**Answer:** $\boxed{a_n = 6n - 2(n-1) = 4n + 2}$

### J.2.4 CASE 4

**Problem.** As shown in the figure, there is a geometric solid composed of a hemisphere and a square pyramid, as shown in the three views. The volume of the solid is ____.

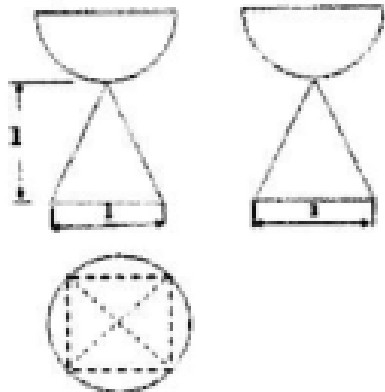

Figure J.20: Solid geometry from three-view drawing.

**Answer:** $\boxed{\dfrac{1}{3} + \dfrac{\sqrt{2}}{6}\pi}$

### J.3 CASE ANALYSIS

Analyzing samples that yield consistently correct or incorrect responses reveals distinct patterns.

**Consistently Correct Cases:** We observe that our approach does not offer a significant advantage for problems characterized by clear images, simple logic, and well-defined results that are easy to express and verify. This is primarily because the visual information is unambiguous, and the solution path is sufficiently straightforward for the base Vision-Language Model. Furthermore, the final answers are often simple integers or fractions, which simplifies verification.

**Consistently Incorrect Cases:** Conversely, for problems involving highly complex image content or those where the results are difficult to describe or verify, achieving a correct answer remains challenging even when reasoning steps are provided. This difficulty arises from the inherent complexity of inferring the correct answer through reasoning alone, which is exacerbated by the difficulty in verifying the intermediate and final results, thereby complicating the optimization task.

## K GENAI USAGE DISCLOSURE

In this work, Large Language Models were utilized to aid in refining and polishing the writing. The author(s) affirm that none of the data analysis, methodology development, or theoretical contributions in this paper involved content generation or research assistance from the GenAI tool. All aspects of the study, including code and data, were developed independently.

## L  LIMITATIONS

Although DIVA-GRPO effectively alleviates reward sparsity and improves performance on multi-modal reasoning tasks, some limitations remain. First, the variant generation process for difficult problems requiring intermediate reasoning steps relies on external models, which may introduce potential biases. Second, difficulty-weighted scaling involves hyperparameters that need careful tuning, which reduces out-of-the-box usability. Finally, DIVA-GRPO has been primarily evaluated on multimodal reasoning tasks, and its generalization to non-multimodal tasks or tasks with fundamentally different reward structures remains to be explored.

