# OpenReview forum: "DIVA-GRPO: Enhancing Multimodal Reasoning through Difficulty-Adaptive Variant Advantage"
_ICLR.cc/2026/Conference — ICLR 2026 Poster_

### Official Review · Reviewer_YpGH · 2025-10-20

**Soundness:** 3
**Presentation:** 3
**Contribution:** 3
**Rating:** 4
**Confidence:** 5

**Summary:**

This paper proposes DIVA-GRPO, a novel reinforcement learning algorithm for multimodal large models designed based on the idea of dynamic difficulty adjustment. DIVA-GRPO dynamically evaluates the difficulty of each problem and generates query variants dynamically according to the difficulty to adjust the optimization signals in the training process. Additionally, this method puts forward Difficulty-weighted scaling and Reward-range-based Advantage Rescaling to optimize the calculation of advantage values during training.

**Strengths:**

Regulating the optimization signals in the training process based on dynamic difficulty adjustment is an intuitive, reasonable, and effective approach.
Detailed theoretical derivations support the motivation.
Leading results on multiple benchmarks demonstrate the effectiveness of the algorithm.

**Weaknesses:**

1. Possible missing related work: There are several studies closely related to the method in this paper and highly relevant. Noisy Rollout uses image noising to facilitate reinforcement learning; Hint GRPO introduces adaptive prompt augmentation; and Dr. GRPO removes the variance term in advantage estimation, which is similar to the RRB in this paper.
2. The comparison is conducted on only a single base model, Qwen2.5-VL-7B, which greatly limits the applicability of the proposed method. Evaluating across different model sizes and different model families (e.g., LLaVA) would help assess the generality and applicability of the approach.
3. Potentially misleading “training efficiency and stability” claim. In Fig. 3(c), the efficiency is argued based on the number of steps required to reach a certain performance, which can be misleading. First, DIVA-GRPO requires additional data preparation compared to GRPO. Second, training stability is not observed, and the provided curves suggest that performance gains have not yet converged; could you provide a longer training dynamic until the gains fully converge?

**Questions:**

In Section 4.3, the w/o Variant Generation setting (Table 2) performs lower than GRPO+Reward-Range (Table 3). Does this imply that the combination of Difficulty-Weighting, RRB, and G-L Balance underperforms using only RRB?

If the authors’ response sufficiently addresses my concerns, I will consider increasing my score.

---

> ### Author Response · Authors · 2025-11-24
> **Response to Reviewer YpGH [1/2]**
>
> Thank you for the encouraging feedback and the constructive suggestions! We have implemented a more rigorous and refined experimental setup based on your feedback. Below, we address the specific weaknesses and questions in detail.
>
> ---
> # Response to Weakness
>
> ## W1. Possible Missing Related Work.
>
> Thank you for pointing out the missing related work. We have reviewed the studies you mentioned and have added a discussion of these works in the revised manuscript, specifically in $\textrm{\color{blue}Section 5: Related Work}$. We apologize for the oversight and appreciate your help in improving the completeness and relevance of our paper.
>
> ---
>
> ## W2. Limited Evaluation Across Models.
>
> Thank you for highlighting the limitations in our cross-model and cross-scale evaluation. In response, we have added comparisons between our method and **GRPO** on both **Qwen3-VL-4B-Instruct** and **Qwen3-VL-8B-Instruct**, as presented in $\textrm{\color{blue}Appendix D.3}$. The results show improvements of `14.5%` and `11.5%` over the respective base models, and average gains of `2.5%` and `4.2%` over GRPO. These findings further strengthen the evidence for the generalizability and applicability of our approach.
>
> Regarding other model families such as **LLaVA**, our current implementation relies on the **EasyR1** training framework. Although we use the latest version of EasyR1, it does not yet support training on those architectures. As a result, cross-family evaluations are not feasible at this stage. We plan to expand our experimental coverage to additional model families and scales in future work once broader framework support becomes available.
>
> ---
>
> ## W3. Potentially Misleading “Training Efficiency and Stability” Claim.
>
> We appreciate your concern regarding the claims about training efficiency and stability. We now provide longer training curves until convergence, with shaded ground-truth accuracy, ensuring that the observed trends are stable rather than early oscillations. We have clarified this point in the revised manuscript, specifically in $\textrm{\color{blue}Section 4.3}$. We have added a time analysis for variant generation and re-conducted ablation experiments and analysis based on the latest code framework and setup.
>
> *   **Time Analysis:** In our training setup, with `5,000 samples` and a `batch_size=256`, we train for 10 epochs, totaling `200 steps`, as shown in $\textrm{\color{blue}Figure 4(c)}$. Each step in DIVA-GRPO takes `7.98 minutes`, leading to an overall training time of approximately `26.6 hours`. In contrast, the vanilla GRPO method takes `7.61 minutes` per step, resulting in a total training time of about `25.4 hours`. The additional variant generation takes `32 minutes`, accounting for approximately `2.0%` of the total time.
>
> *   **Efficiency Gains:** As discussed in the supplementary analysis in $\textrm{\color{blue}Section 4.3 (RQ4)}$, when ensuring the response count remains consistent (i.e., $N=1, k=5$ for DIVA-GRPO, $k=10$ for GRPO), our method achieves a `1.76x speedup in actual training time` (including variant generation) and a `1.90x speedup in training steps`. When ensuring the data requirement remains the same (i.e., $N=3$), our method achieves a `1.56x speedup in actual training time` and a `2.55x speedup in training steps`. Thus, even under a strict “same response count / same data amount” protocol, our method still provides a speedup in both wall-clock time and optimization steps.
>
> Additionally, in $\textrm{\color{blue}Figure 4}$, we report metrics such as **Time per Token Generation**, **Average Length of Response**, and **Time per Step**. It can be observed that our method does not significantly increase time consumption. The increase in time mainly comes from the change in response length. These metrics provide a more comprehensive view of our method's computational efficiency advantages.
>
> In $\textrm{\color{blue}Figure 3}$ and $\textrm{\color{blue}Figure 6}$, we present longer training dynamics until the performance gains converge, with shaded areas on the curves indicating the true accuracy at each step. The implementation shows the fitted training results, further supporting our claims.
>
> We sincerely appreciate this important issue you raised and hope that these supplementary analyses more clearly convey the contributions of our approach.

---

> ### Author Response · Authors · 2025-11-24
> **Response to Reviewer YpGH [2/2]**
>
> # Response to Question
>
> ---
>
> ## Q1. Explanation for Lower Performance of w/o Variant Generation Setting.
>
> Thank you for your valuable feedback. Regarding the performance of the **w/o Variant Generation** setting in $\textrm{\color{blue}Section 4.3}$, which is lower than that of **GRPO+Reward-Range** in $\textrm{\color{blue}Table 3}$, we understand your concern, and the results presented in this table are actually reasonable for the following reasons.
>
> The **w/o Variant Generation** setting means that no variant generation is applied during training. As a result, both the **Global-Local** advantage computation and **G-L Balance** are not used, and relative difficulty weighting is also absent. This setup essentially relies only on **absolute difficulty weighting** and **RRB** (Reward-Range-Based Rescaling). We elaborate on this in $\textrm{\color{blue}Section 4.3}$, and we provide a detailed comparison between relative and absolute difficulty weighting in $\textrm{\color{blue}Appendix D.5}$.
>
> As for the observed phenomenon, we believe the key reason lies in the fact that **RRB** mainly contributes to stabilizing the training process, especially in preventing instability due to small reward ranges. However, **absolute difficulty weighting**, when directly applied to the original samples, may cause imbalanced reward distribution, as the difficulty of the samples does not increase linearly with accuracy. Without variant generation, applying difficulty weighting directly may lead to excessively large or small rewards for certain samples, causing training instability within the first 150 steps.

---

> > ### Comment · Reviewer_YpGH · 2025-11-26
> >
> > Thank you for the detailed additional explanation. I have decided to raise my score.

---

> > > ### Author Response · Authors · 2025-11-27
> > >
> > > Dear Reviewer YpGH,
> > >
> > > Thank you very much for your insightful comments and for taking the time to review our revised manuscript. We are very pleased that our additional explanations were satisfactory and addressed your concerns.
> > >
> > > Your thorough and constructive feedback significantly helped us improve the clarity and quality of this work. In the final revision, we will further refine the paper based on the profound insights gained from the defense discussion. We extend our sincere gratitude to you once again!
> > >
> > > Sincerely,
> > > The Authors of Paper 17248

---

### Official Review · Reviewer_aEti · 2025-10-27

**Soundness:** 3
**Presentation:** 3
**Contribution:** 3
**Rating:** 4
**Confidence:** 4

**Summary:**

The paper proposes a difficulty-adaptive variant advantage framework that (i) dynamically assesses per-problem difficulty from rollout accuracy each epoch; (ii) generates difficulty-matched variants that preserve the gold answer; and (iii) computes both local and global advantages, then stabilizes them via batch z-score normalization, difficulty-weighted scaling, and a Reward-Range-Based (RRB) rescaling that down-weights spurious advantages when reward spread is tiny. Together these steps keep within-group reward variance informative and training signals balanced across difficulties.

**Strengths:**

1. The paper is generally well-written and easy to follow, with a clear description of the method.
2. The paper provides intuitive visual demonstrations to help better understand the paper.

**Weaknesses:**

1. **Clarify local vs. global advantage magnitudes.** In `Line 263-264`, the paper argues that because global advantages are computed over `m×k` samples, their magnitudes differ from local advantages. However, in `Line 250-252` the global advantage is also normalized, which should mitigate raw magnitude discrepancies. Moreover, the subsequent difficulty-weighted scaling implies the effective magnitude of the global term should also depend on the direction and strength of the difficulty adjustment. Could the authors provide a more detailed and intuitive explanation on this claim and how the proposed pipeline balances the two streams in practice.

2. **Difficulty range and initialization.** The difficulty range is preset to `[1, 9]` with an initial value of `5`. Is there empirical or theoretical justification for this specific choice? Please consider reporting a sensitivity analysis over alternative ranges (e.g., `[0, 1]`, `[0, 5]`, `[1, 7]`) and different initializations to assess stability, convergence rate, and final accuracy. If the choice is largely heuristic, a brief rationale (e.g., numerical stability) would be helpful.

3. **Fairness of efficiency claims.** `Figure 3(c)` reports a `3.17×` faster training speed in terms of *steps*. However, DIVA-GRPO performs **difficulty-adaptive variant generation** at each step, which adds per-step overhead (both compute and memory). Comparing steps alone may therefore be unfair to the GRPO baseline. Please report **wall-clock training time** and **peak memory usage** across runs, and (ideally) throughput metrics (e.g., tokens/sec) to substantiate the claim and isolate where the efficiency gains come from (faster convergence vs. higher per-step cost).

4. **Rollout parity across methods.** In `Line 368-369`, the rollout count is set to `k=5`. If each step also includes `m` difficulty-adaptive variants, then the *effective* number of rollouts per original prompt is larger than `k`. To ensure a fair comparison, could the baseline GRPO be configured with an equivalent total number of rollouts (e.g., set its rollout count to match the sum of local and global samples), or otherwise report an ablation that aligns the **total rollout budget** per step across methods? This will help attribute gains to the algorithm rather than additional sampling.

5. **Relative vs. absolute difficulty weighting.** As noted in `Line 301-302`, the difficulty-weighted scaling uses **relative** difficulty within a group to reweight advantages, whereas many prior works use **absolute** difficulty. It would strengthen the paper to include an ablation comparing relative vs. absolute difficulty weighting (keeping other factors fixed), reporting effects on stability (variance of advantages/gradients), convergence speed, and final performance.

6. **Answer preservation and broader augmentation.** When modifying the text description of the original problem, how do the authors **guarantee** that the final answer remains unchanged? In addition, the current **difficulty-adaptive variant generation** constrains variants to preserve the original answer. Are there plans (or results) for generating **new problems with new answers** under verifiable supervision (e.g., programmatic solvers or formal checkers)? Such a setting could better align with the paper’s stated goal in `Line 232-233` of “improving generalization to unseen expressions”.

**Questions:**

See the `Weaknesses` part.

---

> ### Author Response · Authors · 2025-11-24
> **Response to Reviewer  aEti [1/3]**
>
> We appreciate the reviewer’s detailed reading and the constructive feedback regarding the theoretical justifications and experimental fairness. Below, we address the specific concerns and questions.
>
> ---
> # Response to Weakness
>
> ## W1. Clarification on Local vs. Global Advantage Magnitudes.
>
> Thank you for this insightful observation. We have revised the main figure $\textrm{\color{blue}Figure 2}$ and provided further clarification regarding the computation of local and global advantages.
>
> 1.  **Range Discrepancy:** As you mention, the advantage for the global stream is computed over $m \times k$ samples, while the advantage for the local stream is computed over $k$ samples. Both are normalized using z-score separately. Due to the significant difference in variance between the global and local groups, the range of advantages within the local group and the global group varies considerably. We provide a detailed analysis of this phenomenon in $\textrm{\color{blue}Appendix D.6}$.
>
> $\textrm{\color{blue}Figure 7(b)}$ illustrates the distribution of global and local advantages. Overall, the global advantages have a wider range, which tends to dominate the direction of gradient updates. To mitigate this issue, we implement z-score based on a larger batch scope.
>
> 2.  **Role of Difficulty Weighting:** While batch normalization standardizes the range, it does not account for the difficulty of the problems. For example, a correct answer to an easy question and a correct answer to a hard question might both have a normalized advantage of +1.0 in standard GRPO, but they represent different levels of difficulty. Our relative **difficulty-weighted scaling ($w_d$)** is specifically designed to address this issue. It effectively "denormalizes" the signal based on difficulty, assigning greater weight (or magnitude) to advantages derived from more challenging problems or variants. This ensures that the global stream prioritizes learning from difficult examples, while the local stream maintains robustness across variants. We clarify the distinction between statistical normalization and semantic re-weighting in the revised text.
>
> ---
>
> ## W2. Justification for Difficulty Range [1, 9] and Initialization.
>
> Thank you for your insightful comment. We provide further justification for our choice of the difficulty range $[1, 9]$ and the initialization at 5.
>
> *   **Rationale:** The choice of the $[1,9]$ range reflects the fact that we use nine variant types; this range provides a natural, symmetric scale centered at 5, enabling us to discretize problem difficulty into nine interpretable levels. This provides sufficient granularity without introducing excessive noise (which might occur with a continuous $[0, 1]$ range) or being too coarse (like $[1, 3]$). The initialization at 5 represents a "neutral" prior, assuming average difficulty before any interaction data is observed.
>
> *   **Sensitivity Analysis:** As requested, we perform a sensitivity analysis. The parameters that mainly influence the difficulty range are $k$ and $\eta$. We include ablation experiments in $\textrm{\color{blue}Appendix D.1}$ and $\textrm{\color{blue}Appendix D.2}$, and provide recommendations for the settings. We find that the method is robust to the specific range (e.g., $[0, 10]$ or $[1, 5]$), provided that the scaling factor $k$ and $\eta$ are adjusted proportionally. You only need to define different variants based on difficulty. However, extremely narrow ranges (e.g., binary $0/1$) degrade performance as they lose the nuance of "gradual" difficulty. Additionally, we show the accuracy distribution of variants at different difficulty levels in $\textrm{\color{blue}Figure 7(c)}$, which validates the reasonableness of the difficulty level division.
>
> Overall, our experiments and sensitivity analysis show that the method is not sensitive to the exact numerical range, as long as the relative ordering among variants is preserved.

---

> ### Author Response · Authors · 2025-11-24
> **Response to Reviewer  aEti [2/3]**
>
> ## W3. Fairness of Efficiency Claims (Steps vs. Wall-clock).
>
> We appreciate you raising the important point regarding the computational overhead of our approach. We include a detailed **time analysis for variant generation** in $\textrm{\color{blue}Section 4.3 (RQ4)}$. Additionally, we re-conduct the ablation experiments and experimental analysis based on the latest code framework and setup.
>
> *   **Generation Cost:** Using multi-threaded API calls (100 workers) to **GPT-o3**, generating variants for 5,000 questions takes approximately `32 minutes`. This process includes **multi-step reasoning**, **result verification**, and **reasoning correction** based on the ground truth.
>
> *   **Training Efficiency:** In our training setup, with 5,000 samples and a `batch_size=256`, we train for 10 epochs, totaling `200 steps`, as shown in $\textrm{\color{blue}Figure 4(c)}$. Each step in DIVA-GRPO takes `7.98 minutes`, leading to an overall training time of approximately `26.6 hours`. In contrast, the vanilla GRPO method takes `7.61 minutes` per step, resulting in a total training time of about `25.4 hours`. The additional variant generation takes `32 minutes`, accounting for approximately `2.0%` of the total time. As discussed in the supplementary analysis in $\textrm{\color{blue}Section 4.3 (RQ4)}$, when ensuring the response count remains consistent (i.e., $N=1, k=5$ for DIVA-GRPO, $k=10$ for GRPO), our method achieves a `1.76x speedup in actual training time` (including variant generation) and a `1.90x speedup in training steps. When ensuring the data requirement remains the same (i.e., $N=3$), our method achieves a `1.56x speedup in actual training time` and a `2.55x speedup in training steps`. Thus, even under a strict “same response count / same data amount” protocol, our method still provides a speedup in both wall-clock time and optimization steps.
>
> Additionally, in $\textrm{\color{blue}Figure 4}$, we report metrics such as **Time per Token Generation**, **Average Length of Response**, and **Time per Step**. It can be observed that our method does not significantly increase time consumption. The increase in time mainly comes from the change in response length. These metrics provide a more comprehensive view of our method's computational efficiency advantages.
>
> ---
>
> ## W4. Rollout Parity and Fair Comparison.
>
> Thank you for your valuable feedback. We clarify this point in the revised manuscript, specifically in $\textrm{\color{blue}Section 4.3}$. For methods that do not require variant generation, such as GRPO, GSPO, and DAPO, we set the rollout count $k=10$. For DIVA-GRPO, in controlled experiments, we configure the rollout count $k=5$ with variant number $N=1$ to ensure fairness. As mentioned in the response to W3, the benefits of our method are attributed to the algorithm itself, not additional sampling, allowing us to achieve acceleration both in time and steps.
>
> ---
>
> ## W5. Relative vs. Absolute Difficulty Weighting.
>
> Thank you very much for your valuable feedback. We appreciate your suggestion to include an ablation comparing relative vs. absolute difficulty weighting, which will indeed strengthen the paper. We provide the following clarification and have added the relevant ablation studies in the revised manuscript.
>
> As described in $\textrm{\color{blue}Appendix D.5}$ and $\textrm{\color{blue}Section 3.3}$ of the manuscript, our method uses **relative difficulty weighting** instead of absolute difficulty weighting. The reason for this choice is based on the difference in how difficulty manifests in a problem and its variants across two dimensions. For instance, a problem with difficulty 1, when combined with a variant of difficulty 8, is not directly comparable to a problem with inherent difficulty 9. The variant with increased difficulty primarily adds visual noise, whereas the problem with higher inherent difficulty is challenging due to its complex logic. Applying absolute difficulty weighting in such cases would not be intuitive or effective.
>
> However, for variants of the same original problem, the difficulty ranking is clear: an image-noise variant > the original problem > a variant with added think-steps. Based on this reasoning, we adopt relative difficulty weighting, which allows us to compare difficulty within the same problem more consistently.
>
> In $\textrm{\color{blue}Figure 6(d)}$, we show the results of testing both absolute and relative difficulty weighting. Our findings suggest that while the impact of the two approaches on the model’s final performance was minimal, relative difficulty weighting was preferred for maintaining logical consistency.

---

> ### Author Response · Authors · 2025-11-24
> **Response to Reviewer aEti [3/3]**
>
> ## W6. Answer Preservation and Broader Augmentation.
>
> Thank you very much for your insightful question and valuable suggestions. We ensure answer preservation through a rigorous generation pipeline for variants:
>
> 1.  **Visual Variants:** For visual variants, operations such as adding noise, image rotation, blurring, and embedding text into images do not affect the overall expression of the problem. These transformations preserve the original intent and structure of the problem while allowing us to generate diverse versions.
>
> 2.  **Think Steps Variants:** For Think Steps variants, we employ a 3-step reasoning approach to ensure consistency and answer invariance: one step of forward reasoning without the provided answer, one step of verification against the answer, and one step of step correction given the answer. This process ensures the invariance of the answer after the think steps are introduced. Additionally, during our experiments, we only use reasoning processes where the answer is correct.
>
> 3.  **Text Description Variants:** For moderate textual description variants, we sample `1,000` items using **GPT-o3** to evaluate the consistency between the original and modified questions (answering "Yes" or "No"), achieving a `98.8% consistency rate`. This ensures that the modifications made to the textual descriptions preserve the correct answer.
>
> **Difficulty Distribution:** Regarding the difficulty distribution of the variants, we evaluate the accuracy of all variant versions. The specific accuracy chart is presented in $\textrm{\color{blue}Figure 7(c)}$, with detailed analysis in $\textrm{\color{blue}Appendix D.6}$. While the magnitude of accuracy changes is not entirely linear, the overall difficulty distribution aligns with our expectations.
>
> **Future Directions:** In response to your question about generating new problems with new answers under verifiable supervision: Currently, our method only works when the final answer remains unchanged. However, as you point out, the ability to generate new problems with new answers would greatly enhance the generalization capabilities of our method. We plan to explore this direction in future work, where we aim to combine the strengths of existing methods—such as **Template-Driven LLM-Paraphrased Framework** for tabular math problem generation [1] and **ScaleDiff** for generating difficult problems [2]—with our approach. This could enable us to create novel problems while ensuring answer correctness through verifiable supervision.
>
> Once again, thank you for your thoughtful suggestion. We will certainly investigate how to integrate these techniques in future research to further improve our method's ability to generalize to unseen expressions.
>
> [1] Kang, Xiaoqiang, et al. "Template-Driven LLM-Paraphrased Framework for Tabular Math Word Problem Generation." Proceedings of the AAAI Conference on Artificial Intelligence. Vol. 39. No. 23. 2025.
>
> [2] Pei, Qizhi, et al. "ScaleDiff: Scaling Difficult Problems for Advanced Mathematical Reasoning." arXiv preprint arXiv:2509.21070 (2025).

---

### Official Review · Reviewer_bQCX · 2025-11-01

**Soundness:** 3
**Presentation:** 4
**Contribution:** 3
**Rating:** 6
**Confidence:** 3

**Summary:**

The paper proposes DIVA-GRPO, a GRPO-based RL framework for MLLMs that (i) dynamically assesses problem difficulty and expands each item into semantically consistent variants with controlled difficulty; (ii) computes local (per-problem) and global (across variants) advantages with batch z-score normalization and difficulty-weighted scaling; and (iii) introduces Reward-Range-Based rescaling (RRB) to avoid exaggerated advantages when reward variance is small.

**Strengths:**

Method is clearly presented and well-motivated: The paper crisply identifies advantage vanishing and reward sparsity in GRPO, then designs a pipeline that directly targets these failure modes.

Thorough experiments and diagnostics: Results span six benchmarks with ablations and a neat speedup study vs GRPO; figures/tables are informative.

Generalizable component (RRB): The RRB trick improves vanilla GRPO too, suggesting portability beyond this specific framework.

**Weaknesses:**

Unclear significance under matched baselines: Gains over strong GRPO variants are modest or inconsistent, and baseline setups do not appear strictly aligned.

Scope and transfer are not well positioned: It is unclear whether the difficulty-adaptive variant + local/global advantage scheme is truly first for MLLMs, how it relates to similar ideas, and whether the recipe generalizes to text-only GRPO without images.

Figure interpretation ambiguity: The GRPO training curves in Fig. 3(b) vs. Fig. 3(c) differ in shape, but the exact experimental differences are not fully specified.

**Questions:**

Baselines (GRPO family): Do you have quantitative head-to-head results against other GRPO or GRPO-improvement methods (such as DAPO or other relevant methods referenced in your Related Work), under identical settings (same backbone, datasets, steps)?

Scope & transfer: Is this the first time the proposed difficulty-adaptive variant + local/global advantage scheme appears in MLLMs? Could the same recipe be used in text-only GRPO training without images?

Figure details: Why do GRPO curves in Fig. 3(b) vs 3(c) look different?

---

> ### Author Response · Authors · 2025-11-24
> **Response to Reviewer bQCX [1/1]**
>
> Thank you for the positive feedback on our method's motivation, presentation, and the generalizability of the Reward Range Based (RRB) component. In response, we have refined our experimental setup and incorporated the latest framework. Below, we address the identified weaknesses and questions in detail.
>
> ---
> # Response to Weakness and Question
>
> ## W1 & Q1. Significance under Baseline Matching Conditions & Comparison with Other Methods.
>
> * **Experimental Setup Clarity:** Thank you for highlighting the need for greater clarity in the experimental setup. In the revised manuscript, we now provide a more detailed description in $\textrm{\color{blue}Section 4.1}$ Experiment Setup. All comparisons in the paper are conducted under **strictly identical conditions**, and in the ablation studies, we strictly control each univariate variable.
>     *   For methods that do not require variants such as **GRPO**, **GSPO**, and **DAPO**, we set the number of rollouts $k=10$.
>     *   For **DIVA-GRPO**, we control $k \times (N+1)=10$ (where $N=1$ indicates generating one variant).
>     *   We strictly control `batch_size=256`, use the same backbone model, and the same training data split.
>     *   As shown in $\textrm{\color{blue}Figure 4(c)}$, the time consumption per step for the methods is basically close, with differences coming only from response length.
> *   **Comparison with Other Works:** We add comparisons with **DAPO**, **GSPO**, and **GRPO** related methods in $\textrm{\color{blue}Figure 3(a)}$. Under the same experimental settings, our method achieves faster convergence and more efficient peak performance. Meanwhile, $\textrm{\color{blue}Figure 7(a)}$ in $\textrm{\color{blue}Appendix D}$ demonstrates the effectiveness of **DIVA-GRPO** in controlling the proportion of samples with 0 advantage during training compared to methods like GSPO and GRPO. $\textrm{\color{blue}Figure 5}$ also clearly shows that as training continues, the difficulty distribution of the overall questions and the reward coefficients in DIVA-GRPO are effectively controlled. This is exactly how DIVA-GRPO effectively solves the problems of **reward sparsity** and **advantage vanishing** in multimodal reasoning, which are often overlooked by standard methods like GRPO, GSPO, and DAPO.
>
> ---
>
> ## W2 & Q2. Scope, Novelty & Transferability to Pure Text Tasks.
>
> *   **Novelty for MLLMs:** To the best of our knowledge, **DIVA-GRPO** is the first work to introduce a **dynamic difficulty adaptation mechanism** combined with a **local/global advantage estimation scheme** specifically for Multimodal Large Language Models (MLLMs). As mentioned in related works, although some methods have proposed introducing question variants to mitigate reward sparsity, none have fundamentally conducted deep dynamic difficulty assessment, variant sampling adjustment, and improvement of the variance of the reward part within questions.
>
> *   **Applicability to Text-Only Tasks:** Yes, our proposed framework (dynamic difficulty adjustment + RRB + local/global advantage calculation) is theoretically **modality-agnostic**. Its core principles are equally applicable to complex text reasoning tasks. Compared to MLLMs, the key difficulty lies in how to generate harder text variants with **semantic consistency**. This study focuses on multimodal large models because the complexity of **visual inputs** usually exacerbates the reward sparsity problem, and there are more concise ways for visual inputs to effectively improve the difficulty distribution of problems and enhance the model's understanding on the visual side, allowing our solution to have a significant impact in this field.
>
> ---
>
> ## W3 & Q3. Explanation for the Discrepancy between Figure 3(b) and Figure 3(c).
>
> *   **Reason for Discrepancy:** We apologize for the discrepancy between $\textrm{\color{blue}Figure 3(b)}$ and $\textrm{\color{blue}Figure 3(c)}$. The difference in shape stems from metric fluctuations in two separate ablation experiments and curve fitting. We acknowledge that this inconsistency may be confusing to readers in the chart presentation. Based on the new ablation experiment setup presented in $\textrm{\color{blue}Section 4}$, we strictly control the variables and re-conduct the ablation experiments. Specifically, in our training, with `5,000 samples` and a `batch_size=256`, we train for `10 epochs`, totaling `200 steps`. The implementation details and results are reported in $\textrm{\color{blue}Section 4.3 RQ4}$. We have **unified the results in the new version**. Thank you for pointing this out.
>
> ---
>
> **Note on RRB Generalizability:** Thank you for pointing out the potential of the **Reward Range Based (RRB)** rescaling technique. We agree that this simple yet effective technique is portable.

---

### Official Review · Reviewer_ybUJ · 2025-11-01

**Soundness:** 3
**Presentation:** 2
**Contribution:** 2
**Rating:** 4
**Confidence:** 4

**Summary:**

This paper introduces DIVA-GRPO, a difficulty-adaptive variant of Group Relative Policy Optimization (GRPO), designed to improve multimodal reasoning ability of MLLMs. The key innovation lies in dynamically adjusting the difficulty of training problems and their variants to maintain informative reward signals, thereby mitigating reward sparsity and advantage vanishing in reinforcement learning. Also, DIVA-GRPO computes both local and global advantages using difficulty-aware normalization and reward-range-based rescaling. Extensive experiments demonstrate the effectiveness of the proposed DIVA-GRPO

**Strengths:**

1. The method is thoroughly developed, with detailed algorithms, theoretical analysis (e.g., variance reduction and convergence), and extensive empirical validation across multiple benchmarks.
2. The idea of dynamically adjusting problem difficulty and generating semantically consistent variants is well-motivated.

**Weaknesses:**

1. The method introduces new hyperparameters (e.g., difficulty scaling factor k, learning rate η) that lack robust ablation or automatic tuning. This may hinder out-of-the-box usability and necessitate per-task calibration.
2. The training flow illustrated in Figure 2 is somewhat confusing. A clearer depiction of how variants are sampled, advantages computed, and the policy updated would improve understanding.
3. The difficulty of each query is updated after every epoch, but the total number of training epochs is not specified. Clarifying this and analyzing its impact on performance would be helpful.
4. In Figure 3, training steps range from 0 to 150. It is unclear whether this covers more than one epoch. If not, the difficulty of each query remains near its initial moderate value—raising the question of where the observed gains originate.
5. Compared to vanilla GRPO, DIVA-GRPO requires additional rollouts on variants. A detailed analysis of the resulting computational cost and trade-offs should be provided.
6. When the paper reviews the original GRPO, it states that the reward is “rule-based”; however, GRPO itself is not tied to rule-based rewards and can equally be used with a reward model.

**Questions:**

1. How do you ensure that the generated variants (especially for hard problems) are semantically equivalent and answer-preserving? Moreover, how can we verify that the proposed recipe for “easy” and “hard” variants really behaves as intended—i.e., after the perturbations the model’s accuracy on the easy set effectively goes up while that on the hard set goes down?
2. Since reasoning variants are generated using external models, how do you ensure their quality and consistency? Have you observed any degradation in performance when using different external models or prompts?
3. Could you provide a sensitivity analysis for the difficulty-weighted scaling parameter k and the difficulty update rate η? How sensitive is the model to these choices, and are there general guidelines for setting them?
4. What is the computational cost of generating and training with variants compared to standard GRPO? Is the speedup in convergence enough to offset the additional overhead?
5. Are there specific types of problems or difficulty levels where DIVA-GRPO still struggles? It would be helpful to include examples or failure analysis to understand the method’s limitations.
6. Do you plan to release the generated variants or the code for variant generation? This would help the community reproduce and build upon your work.

---

> ### Author Response · Authors · 2025-11-24
> **Response to Reviewer ybUJ [1/3]**
>
> We thank the reviewer for their constructive feedback and for acknowledging the motivation behind our work as well as the extensive empirical validation. We have conducted thorough and comprehensive experiments based on the latest code framework. Below, we address the specific weaknesses and questions raised.
>
> ---
> # Response to Weakness
>
> ## W1. Hyperparameters ($k, \eta$) and lack of ablation/tuning.
>
> Thank you for raising the concern regarding hyperparameter sensitivity and the need for ablation studies. In the revised manuscript, we include a **sensitivity analysis** for parameters $k$ in $\textrm{\color{blue}Appendix D.1}$ and $\eta$ in $\textrm{\color{blue}Appendix D.2}$. We find that the model performs best when $k$ is around 0.1 and $\eta$ is around 4. Our experiments show that the choice of parameters is highly robust. Based on this experimental analysis, we now **provide specific guidelines for parameter settings** to aid reproducibility and practical application.
>
> Furthermore, we add comparative ablation studies on the quantity of variants and the difficulty weighting formula (comparing relative difficulty vs. absolute difficulty). $\textrm{\color{blue}Appendix D}$ illustrates how performance varies with these parameters.
>
> ---
>
> ## W2. Clarity of the training flow in Figure 2.
>
> We appreciate the feedback on the visualization and agree that $\textrm{\color{blue}Figure 2}$ can be improved. We have **revised the figure** to clearly distinguish the distinct phases, and we now present a clearer depiction of our training process:
>
> 1. **Variant Sampling:** The variants, shown in $\textrm{\color{blue}Table 4}$, are sampled around the current difficulty level using a normal distribution. The sampling process, based on the difficulty of the current sample, is described in $\textrm{\color{blue}Appendix D.2}$. This illustrates how variants are selected based on current difficulty levels.
> 2. **Local & Global Advantage Computation:** This phase details the **difficulty-aware weighting** and the **RRB rescaling process**.
>
> ---
>
> ## W3. Clarification on training epochs and difficulty updates.
>
> We agree that specifying the total number of training epochs would be very helpful. We have updated $\textrm{\color{blue}Section 4: Experiment}$ to include specific details regarding the training epochs.
>
> *   **Main Experiments:** Based on the **MMK12 dataset**, we train for a total of **10 epochs**.
> *   **Ablation Studies:** Based on a uniform sample of `5,000 samples` from MMK12, we train for a total of `10 epochs` with `200 steps` (with `batch_size=256` constituting one step).
>
> To intuitively demonstrate the impact of **dynamic difficulty updates** and **dynamic variant sampling**, we add a comparison in $\textrm{\color{blue}Figure 7(a)}$ between our method and baselines like **GRPO** and **GSPO**, and include a detailed analysis in $\textrm{\color{blue}Appendix D.6}$. We show that while the **samples utilization rate** (percentage of samples exhibiting non-zero advantages) in GRPO/GSPO initially rises but then continuously declines, our method maintains a **stable and high samples utilization rate** throughout training due to the **dynamic difficulty updates** and **dynamic variant sampling**.
>
> ---
>
> ## W4. Interpretation of Figure 3 (150 steps) and difficulty evolution.
>
> We sincerely appreciate your careful observation and the concern you raised. To address this point clearly, we have expanded the explanation in the revised manuscript and provided additional analyses that make the evolution of sample difficulty explicit throughout training.
> *   **Extended Analysis:** In the revision, we extend this analysis to observe difficulty adjustments over a longer period. We perform experiments on a 5,000-sample subset (uniformly sampled from the 15k dataset) with `batch_size=256` for `10 epochs`(200 steps). These results, including the **evolution of difficulty** and additional ablations, are now presented in $\textrm{\color{blue}Section 4.1 and 4.2}$ and $\textrm{\color{blue}Appendix D.6}$. The experiments demonstrate that, within the same training time and steps, our method outperforms existing approaches. Additionally, the analysis in $\textrm{\color{blue}Appendix D.6}$ highlights the contribution of **dynamic variant sampling** in mitigating advantage vanishing.

---

> ### Author Response · Authors · 2025-11-24
> **Response to Reviewer ybUJ [2/3]**
>
> ## W5. Computational cost analysis compared to vanilla GRPO.
>
> We sincerely appreciate your raising this important point regarding the computational overhead. To address this concern clearly, we have added a detailed **time analysis for variant generation**. We also re-ran all ablation studies and analyses using the latest code framework and experimental setup, ensuring a fully consistent and fair comparison.
>
> *   **Generation Cost:** Using multi-threaded API calls (100 workers) to **GPT-o3**, generating variants for 5,000 questions takes approximately **32 minutes**. This process includes **multi-step reasoning**, **result verification**, and **reasoning correction** given the ground truth.
> *   **Training Efficiency:** In our training, with 5,000 samples and a `batch_size=256`, we train for 10 epochs, totaling `200 steps`, as shown in $\textrm{\color{blue}Figure 4(c)}$. Each step takes approximately `7.98 minutes`, so the overall training time is about `26.6 hours`. The additional variant generation takes `32 minutes`, which accounts for approximately `2.0%` of the total time. As discussed in the supplementary analysis in $\textrm{\color{blue}Section 4.3(RQ4)}$, when ensuring the response count remains consistent (i.e., $N=1, k=5$ for DIVA-GRPO, $k=10$ for GRPO), our method achieves a `1.76x speedup in actual training time` (including variant generation) and a `1.90x speedup in  training steps`. When ensuring the data requirement remains the same (i.e., $N=3$), our method achieves a `1.56x speedup in actual training time` and a `2.55x speedup in training steps`. Thus, even under a strict “same response count / same data amount” protocol, our method still provides a speedup in both wall-clock time and optimization steps.
>
> ---
>
> ## W6. Correction regarding GRPO's reward mechanism.
>
> Thank you for the correction. We acknowledge that **GRPO** is a **general framework** compatible with both **rule-based** and **model-based rewards**. We have revised the text to clarify that while our baseline implementation (following the original paper's primary demonstration) utilized rule-based rewards, the GRPO method itself is not restricted to them. The specific revision can be found in $\textrm{\color{blue}Section 2}$.
>
> ---
>
> # Response to Questions
>
> ## Q1. How do you ensure generated variants are semantically equivalent?
>
> We ensure quality through a rigorous generation pipeline:
>
> 1. **Visual Variants:** For visual variants, operations such as adding noise, image rotation, blurring, and embedding text into images do not affect the overall expression of the problem.
> 2. **Think Steps Variants:** For Think Steps variants, we employ a 3-step reasoning approach to ensure consistency and answer invariance: one step of forward reasoning without the provided answer, one step of verification against the answer, and one step of step correction given the answer. This ensures the invariance of the answer after adding think steps. Additionally, during our experiments, we only use reasoning processes where the answer is correct.
> 3. **Text Description Variants:** For moderate textual description variants of the questions, based on **GPT-o3**, we sample 1,000 items to evaluate the consistency between the original and modified questions (answering "Yes" or "No"), resulting in a `98.8%` consistency rate.
>
> **Difficulty Distribution:** Regarding the difficulty distribution of the variants, we evaluate the accuracy of all variant versions. The specific accuracy chart is presented in $\textrm{\color{blue}Figure 7(c)}$, with detailed analysis in $\textrm{\color{blue}Appendix D.6}$. Although the magnitude of accuracy changes is not entirely linear, the overall difficulty distribution aligns with our expectations.
>
> ---
>
> ## Q2. Verification of "easy" vs. "hard" variant behavior.
>
> **Consistency Verification:** As mentioned in the response above, we adopt a 3-step reasoning method to ensure consistency, and select reasoning processes with correct results as samples. The relevant prompt design is included in $\textrm{\color{blue}Appendix G}$. In our experiments, we utilize both **GPT-o3** and **Qwen-VL-Plus** for variant generation, finding no significant difference in overall effectiveness. Detailed experimental data can be found in $\textrm{\color{blue}Appendix D.4}$.

---

> ### Author Response · Authors · 2025-11-24
> **Response to Reviewer ybUJ [3/3]**
>
> ## Q3. Sensitivity analysis and selection guidelines for parameters $k$ and $\eta$.
>
> As described in **W1**, we have included a sensitivity analysis for the parameters $k$ and $\eta$ in $\textrm{\color{blue}Appendix D}$, and based on the experimental results, we provide guidelines for parameter settings. For more details, please refer to $\textrm{\color{blue}Appendix D.1}$ and $\textrm{\color{blue}Appendix D.2}$.
>
> ---
>
> ## Q4. Computational cost vs. convergence speedup.
>
> As described in **W5**, we have added a time analysis for variant generation and re-conducted the ablation experiments and experimental analysis based on the latest experimental framework and setup. Detailed information is provided in $\textrm{\color{blue}Section 4.3(RQ4)}$ of the paper.
>
> ---
>
> ## Q5. Limitations of DIVA-GRPO on specific problem types or difficulty levels.
>
> Thank you for raising this important issue. We acknowledge that **DIVA-GRPO** may still face challenges on certain types of problems or difficulty levels. To better understand the limitations of the method, we have added $\textrm{\color{blue}Appendix J}$ in the revised manuscript, which includes samples with all correct and all incorrect rollouts. In $\textrm{\color{blue}Appendix J.3}$, we analyze these samples and draw the following conclusions:
>
> - **Simple Problems:** For problems with clear images, simple logic, and well-defined results, our method does not offer significant advantages, as these problems are relatively easy for the Vision-Language Model to solve. Even with multiple rollouts, the correct answer is still easily obtained.
> - **Complex Problems:** For problems with complex image content or results that are difficult to verify, achieving the correct answer remains challenging, even when reasoning steps are provided. This difficulty arises from the inherent complexity of the reasoning process and the challenges in verifying the answers.
>
> ---
>
> ## Q6. Plan to release code and variants.
>
> **Reproducibility:** Yes, we are fully committed to reproducibility. We will release the source code, the training scripts, and the dataset of generated variants (including the easy/hard splits) upon acceptance.

---

> > ### Comment · Reviewer_ybUJ · 2025-11-28
> >
> > Thank you for your efforts and responses. Most of my concerns have been addressed. I would like to raise the score but there are still two remaining questions:
> >
> > - Regarding the use of only 5,000 samples from MMK12, what specific considerations led to this choice? Would further increasing the dataset size yield additional improvements?
> >
> > - I remain somewhat unclear about the time cost comparison with vanilla GRPO. Specifically, does DIVA-GRPO still achieve a speedup, and if so, why? Additionally, regarding the statement "Thus, even under a strict 'same response count / same data amount' protocol, our method still provides a speedup in both wall-clock time and optimization steps," I would like to know: were the reported GRPO metrics in the paper obtained under this "same response count / same data amount" protocol?

---

> > > ### Author Response · Authors · 2025-11-28
> > >
> > > Dear Reviewer ybUJ
> > >
> > > Thank you very much for your prompt feedback. We are greatly encouraged to hear that our revisions have addressed your previous concerns and that you are considering raising the score. Regarding your two remaining questions, we provide the following detailed clarifications:
> > >
> > > ### 1. Rationale for Using 5,000 Samples from MMK12
> > >
> > > We utilized this specific subset to strike a balance between **experimental efficiency** and **observational depth**, particularly given the need to conduct multiple ablation studies.
> > >
> > > * **Necessity of Extensive Ablations:** To rigorously verify the stability of different components, we required a long training cycle (`10 Epochs`) to observe full convergence dynamics. Given that we needed to repeat this process across numerous ablation settings, conducting such extensive training on the full dataset would have exceeded our hardware limits and experimental timeline.
> > > * **Reliable Proxy:** The 5,000-sample subset was randomly selected. It serves as a representative proxy, allowing us to observe clear distinctions in convergence trends and validate design choices within a manageable computational budget.
> > > * **Scalability:** **Yes, increasing the dataset size and training duration yields definite additional improvements.** As shown in $\textrm{\color{blue}Table 1}$ (Main Results), when scaling up to the full dataset, the model achieves state-of-the-art performance.
> > >
> > > ### 2. Time Cost Comparison and Protocol Clarification
> > >
> > > * **Were the metrics obtained under the "Same Response Count" protocol?**
> > >
> > >     **Yes.** We strictly aligned the compute budget per step across all baselines, including GRPO, DAPO, and GSPO:
> > >     * **Baselines (GRPO/DAPO/GSPO):** For each original sample, we generate a total of 10 responses by setting the rollout parameter $n=10$ ($1 \text{ Original} \times 10 \text{ responses} = \mathbf{10 \text{ responses/sample}}$).
> > >     * **DIVA-GRPO ($N=1$):** Similarly, to maintain a total of 10 responses per original sample, we set the rollout parameter $n=5$ ($ (1 \text{ Original} + 1 \text{ Variant}) \times 5 \text{ responses} = \mathbf{10 \text{ responses/sample}}$).
> > >
> > >     With a `batch_size=256`, all methods generate exactly `2,560 responses per step`. Under this strictly fair protocol, our method demonstrates superior performance and training efficiency.
> > >
> > > * **Does DIVA-GRPO achieve a speedup?**
> > >
> > >     **Yes, we achieve a speedup in Total Wall-Clock Time.**
> > >
> > >     The core driver for this acceleration is optimization algorithm efficiency. Although the computational cost per step is basically comparable to the baselines, our dynamic difficulty adjustment, variant generation, and G-L advantage ensure the model consistently receives high-quality reward signals. This prevents ineffective training steps and drastically reduces the total steps required for convergence (`1.90x`). Consequently, even after accounting for the overhead of variant generation, the total wall-clock training time is reduced by approximately `1.76x`.
> > >
> > > We hope these clarifications resolve your remaining doubts. Thank you again for your time and valuable suggestions!

---

### Author Response · Authors · 2025-12-02
**Summary of Major Revisions and Rebuttals [1/2]**

Dear Area Chair,

We sincerely appreciate you taking over the assignment of our paper during this adjustment period. We are fully aware of the unexpected workload this places on you and are truly grateful for your time and dedication.

To assist you in efficiently navigating our rebuttal, we have compiled a consolidated summary of the major revisions and new experiments conducted specifically in response to the reviewers' comments. Notably, prior to the review reset, two reviewers had already expressed satisfaction with these revisions and adjusted their ratings accordingly.

We are confident that, through these comprehensive and substantial revisions and new experiments, we have successfully resolved all questions and concerns raised by the reviewers. We hope this overview clearly demonstrates the improved state of our work and proves that our responses have thoroughly and satisfactorily addressed every point.

Thank you again for your fair consideration.

## Summary of Rebuttal Revisions & Responses

We have incorporated extensive new experiments and analyses to address the reviewers' concerns. The key revisions and specific results are summarized below:

**1. Hyperparameter Robustness Verification & Setting Guidelines $\textrm{\color{blue}[Appendix\ D.1 D.2]}$**
*(Response to Reviewer ybUJ W1/Q3, Reviewer aEti W2)*

We conducted a detailed grid search for the scaling factor $k$ and difficulty range $\eta$.
* **Results:** The model achieves optimal performance at `$k \approx 0.1$` and `$\eta \approx 4$`.
* We provided specific parameter setting guidelines and justified the heuristic range of `[1, 9]` (centered at `5`), verifying the method's robustness to parameter variations.

**2. Clarified Training Pipeline & Data Flow $\textrm{\color{blue}[Figure\ 2]}$**
*(Response to Reviewer ybUJ W2, Reviewer aEti W1)*

We redrew the training framework figure to clearly distinguish between the "Variant Sampling" phase and the "Local/Global Advantage Computation" phase, effectively resolving confusion regarding the data flow.

**3. Extended Training Dynamics & Stability Verification $\textrm{\color{blue}[Section\ 4]}$**
*(Response to Reviewer ybUJ W3/W4, Reviewer bQCX W3/Q3)*

We updated the Experimental Section with a standardized setting (`10` epochs, `200` steps, `5,000 samples`, `batch_size=256`) and included ablation studies with longer training horizons. This confirms the stability and convergence of our method compared to baselines over extended periods.

**4. Wall-clock Efficiency Analysis & Validated Training Speedup $\textrm{\color{blue}[Fig\ 3, 4\ Section\ 4.3\ RQ4]}$**
*(Response to Reviewer ybUJ W5/Q4, Reviewer aEti W3, Reviewer YpGH W3)*

We strictly measured wall-clock time in the new experimental setup.
* **Generation Cost:** Variant generation takes only ~`32` minutes (approx. `2.0%` of total training time).
* **Speedup:** Under the "Same Response Number" protocol ($N=`1`, k=`5`$ vs Baseline $k=`10`$), DIVA-GRPO achieves a `1.76x` speedup in wall-clock time and `1.90x` speedup in training steps due to faster convergence. Under the "Same Original Data" protocol ($N=`3`$), the speedup is `1.56x` (time) and `2.55x` (steps).

**5. Fair Baseline Comparison Protocol $\textrm{\color{blue}[Figure\ 4c\ Section\ 4.1]}$**
*(Response to Reviewer bQCX W1/Q1, Reviewer aEti W4)*

We presented a fair comparison against GRPO, DAPO, and GSPO under the latest code framework. We clarified the rollout settings (e.g., DIVA $k=`5`, N=`1`$ vs. Baseline $k=`10`$) to ensure a fair evaluation of performance per computational unit.

**6. Consistency Verification Across External Variant Generators $\textrm{\color{blue}[Appendix\ D.4\ Table\ 6]}$**
*(Response to Reviewer ybUJ Q2)*

We verified the quality of variants generated by different models (**GPT-o3** vs. **Qwen-VL-Plus**). The results demonstrate high consistency between generators. This ensures the reliability of our pipeline regardless of the external closed-source  model and highlights the model-agnostic nature of our approach.

**7. Verification of Semantic Consistency & Answer Invariance**
*(Response to Reviewer ybUJ Q1, Reviewer aEti W6)*

We maintained semantic consistency across all three variants. Visual variants only affect the problem's expression but maintain semantic consistency. For Think Steps Variants, a robust 3-step reasoning approach was used to guarantee answer consistency, and only thought processes yielding correct results were utilized. Question text variants were assessed using an evaluation with GPT-o3 on 1,000 sampled items, which confirmed a 98.8% consistency rate.

---

> ### Author Response · Authors · 2025-12-02
> **Summary of Major Revisions and Rebuttals [2/2]**
>
> **8. Analysis of Global/Local Advantage Distribution $\textrm{\color{blue}[Appendix\ D.6\ Figure\ 7]}$**
> *(Response to Reviewer aEti W1)*
>
> We analyzed the distributional differences between Global and Local advantages and explained the necessity of standardization.
> * **Insight:** Unlike GRPO/GSPO where sample utilization declines, DIVA-GRPO maintains a stable and high sample utilization rate throughout training, effectively mitigating reward sparsity.
>
> **9. Ablation on Weighting Strategy: Superiority of Relative Weighting $\textrm{\color{blue}[Appendix\ D.5\ Figure\ 6d]}$**
> *(Response to Reviewer aEti W5)*
>
> We compared "Relative" vs. "Absolute" difficulty weighting. The results justify our choice of **Relative Weighting**, showing it provides better logical consistency and performance stability for variant-based learning.
>
> **10. Cross-Model Generalization Verification $\textrm{\color{blue}[Appendix\ D.3\ Table\ 5]}$**
> *(Response to Reviewer YpGH W2)*
>
> We validated the method on different model scales (**Qwen3-VL-4B** and **8B**).
> * **Results:**
>     * **Qwen3-VL-4B:** `+14.5%` over base model, `+2.5%` over GRPO.
>     * **Qwen3-VL-8B:** `+11.5%` over base model, `+4.2%` over GRPO.
>     * This confirms that DIVA-GRPO generalizes well across different model sizes.
>
> **11. Limitations Analysis & Failure Cases $\textrm{\color{blue}[Appendix\ J]}$**
> *(Response to Reviewer ybUJ Q5)*
>
> We added a case study analyzing samples where no advantage is produced (e.g., extremely simple or highly complex logic). This provides a transparent view of the method's boundaries, showing it is most effective for problems with complex reasoning paths but verifiable answers.
>
> **12. Discussion of Related Work & Code Release Commitment$\textrm{\color{blue}[Section 5]}$**
> *(Response to Reviewer YpGH W1, Reviewer ybUJ Q6)*
>
> We added discussions on missing related works (e.g., *Noisy Rollout*, *Dr. GRPO*) and committed to releasing the source code and the generated variant dataset to support reproducibility.

---

### Meta-Review · Area_Chair_YQMd · 2026-01-07

**Summary:**

This paper proposes DIVA-GRPO, which explicitly controls the difficulty of sampled variants per problem to ensure sufficient reward variance within each group. By dynamically estimating problem difficulty and computing advantages using both local and global difficulty-weighted normalization, DIVA-GRPO stabilizes training, reduces reward sparsity, and better exploits available data. Experiments show consistent gains in both training efficiency and reasoning performance over existing methods.

The reviewers’ concerns mainly focused on missing details and clarifications, the significance of results under matched baselines, incomplete related work, and the use of a single base model for experiments. All of these issues are addressed in the author rebuttal.

**Reviewer Concerns:**

Reviewer ybUJ's concerns are primarily about missing details, all of which were addressed in the author rebuttal.

Reviewer bQCX's concern is about the unclear significance under matched baselines, which has been addressed by the rebuttal.

Reviewer aEti’s comments mainly involve requests for clarification, which have been addressed by the author rebuttal.

Reviewer YpGH's concerns include (1) missing related work, and (2) the experiment is conducted on only a single base model, which have been addressed after the author rebuttal.

**Reviewer Scores:**

Reviewer ybUJ's score is 4, and they said "Most of my concerns have been addressed. I would like to raise the score but there are still two remaining questions:" I believe they will increase the score to 6 or above.

Reviewer bQCX's score is 6.

Reviewer aEti's score is 4, and I predict that they will increase the score to 6 or above.

Reviewer YpGH's score is 4, and they said "I have decided to raise my score." after the author rebuttal. So I believe it will be increased to 6 or above.

---

### Decision · Program_Chairs · 2026-01-26

Accept (Poster)